# INCLUSIVEVIDPOSE: BRIDGING THE POSE ESTIMATION GAP FOR INDIVIDUALS WITH LIMB DEFICIENCIES IN VIDEOS

**Heming Du,**[*] **Jiaying Ying,**[*] **Sen Wang & Xue Li**
School of Electrical Engineering and Computer Science
The University of Queensland

**Kaihao Zhang**
College of Engineering and Computer Science
The Australian National University

**Xin Yu**[†]
Australian Institute for Machine Learning
The University of Adelaide

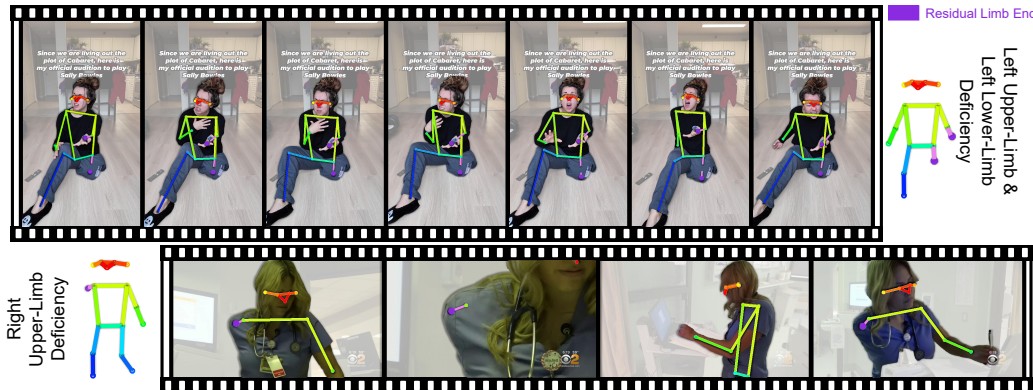

Figure 1: **Demonstration of InclusiveVidPose Dataset.** The top row depicts a subject with left upper and lower limb deficiency across frames, and the bottom row depicts a subject with right upper limb deficiency. Unlike existing pose estimation datasets, we focus on human pose estimation for individuals with limb deficiencies and add custom residual limb end keypoints highlighted in purple to capture anatomical variations. All the displayed keypoints are manually annotated.

## ABSTRACT

Approximately 445.2 million individuals worldwide are living with traumatic amputations, and an estimated 31.64 million children aged 0–14 have congenital limb differences, yet they remain largely underrepresented in human pose estimation (HPE) research. Accurate HPE could significantly benefit this population in applications, such as rehabilitation monitoring and health assessment. However, the existing HPE datasets and methods assume that humans possess a full complement of upper and lower extremities and fail to model missing or altered limbs. As a result, people with limb deficiencies remain largely underrepresented, and current models cannot generalize to their unique anatomies or predict absent joints. To bridge this gap, we introduce **InclusiveVidPose** Dataset, the first video-based large-scale HPE dataset specific for individuals with limb deficiencies. We collect 313 videos, totaling 327k frames, and covering nearly 400 individuals with amputations, congenital limb differences, and prosthetic limbs. We adopt 8 extra keypoints at each residual limb end to capture individual anatomical variations. Under the guidance of an internationally accredited para-athletics classifier, we annotate each frame with pose keypoints, segmentation masks, bounding boxes, tracking IDs, and per-limb prosthesis status. Experiments on InclusiveVidPose highlight the limitations of the existing HPE models for individuals with limb

---

[*]Equal contribution.
[†]Corresponding Author

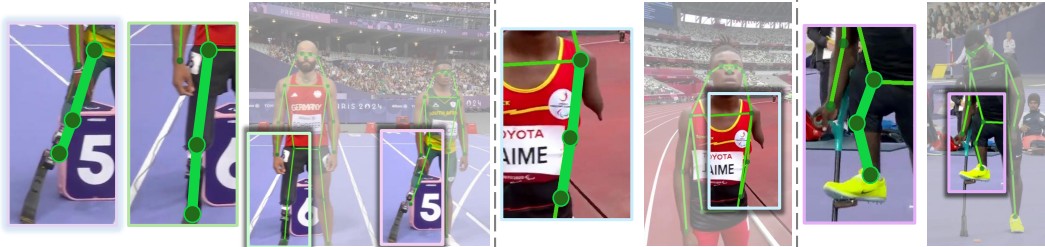

Figure 2: **Examples of keypoint predictions by the ViTPose base model trained the COCO dataset.** Left: prostheses are erroneously predicted as natural ankles, leading to false right-ankle detections. Middle: the model fails to localize the residual limb end and places the left-wrist keypoint on the torso. Right: pronounced asymmetry in thigh length results in the right-knee keypoint being placed at an anatomically implausible midpoint between hip and ankle. These cases show limited generalization to limb differences.

deficiencies. We introduce a new evaluation metric, Limb-specific Confidence Consistency (LiCC), which assesses the consistency of pose estimations between residual and intact limb keypoints. We also provide a rigorous benchmark for evaluating inclusive and robust pose estimation algorithms, demonstrating that our dataset poses significant challenges. We hope InclusiveVidPose spur research toward methods that fairly and accurately serve all body types. The project website is available at: ○ InclusiveVidPose.

# 1 INTRODUCTION

According to the Global Burden of Disease (GBD) 2021 report (Institute for Health Metrics and Evaluation (IHME), 2024), approximately 445.2 million individuals worldwide are living with traumatic amputations, and an estimated 31.64 million children aged 0–14 have congenital limb differences. These people could benefit greatly from accurate human pose estimation (HPE) in applications such as rehabilitation monitoring and health assessment. Regrettably, existing HPE systems are trained on the existing datasets and are not designed for atypical anatomies or prosthetic occlusions, leaving this population unsupported.

Widely used benchmarks like MS COCO (Lin et al., 2014b) and MPII Human Pose (Andriluka et al., 2014) include only able-bodied people with complete sets of keypoints. These datasets and the methods built on them assume that every keypoint of a presented individual exists, making no provision for missing or altered limbs. As demonstrated in Figure 2, the MS COCO trained ViTPose model produces significant errors when applied to images of individuals with limb deficiencies. On the basis of these observations, we find that people with limb deficiencies are excluded from current research, and models trained on the existing datasets fail to generalize to their anatomies.

To address this gap, we introduce InclusiveVidPose, the first video-based HPE dataset focused on individuals with limb deficiencies. We collect 313 videos, totaling 327k frames, from 398 participants who have amputations, congenital limb differences, and prosthetic limbs. Developing such a dataset presents unique challenges. As illustrated in Figures 1, impairments vary widely across individuals, including differences in the appearance and length of residual limbs, their anatomical location, and the effects of different types of prostheses. In the existing keypoint schema, there is no predefined sequence that accommodates these anatomical variations. To more accurately represent these cases, we adopt additional keypoints at the ends of residual limbs.

Unlike single-image datasets, InclusiveVidPose leverages video context to disambiguate occluded limbs from truly absent ones, ensuring precise residual-limb end annotations across frames. We ensure the annotation quality through guidance from an internationally accredited para-athletics classifier and verification by multiple trained annotators. For each frame, we provide standard and residual-limb keypoints, segmentation masks, subject tracking IDs, bounding boxes, and the prosthesis status of each residual limb.

Table 1: **Comparison of existing datasets for human pose estimation.** InclusiveVidPose offers unique and richer annotations than previous datasets. It is also the first pose estimation dataset to focus on individuals with limb deficiencies and to include keypoints at the ends of residual limbs.

| Dataset | #Image/ Frames | #Pose | Segmentation Mask | Bounding Box | Tracking ID | Individuals with Limb Deficiencies | Residual Limb End Annotations | Prosthesis Info |
|---|---|---|---|---|---|---|---|---|
| MPII Human Pose Andriluka et al. (2014) | 25k | 40k | | | | | | |
| MS COCO Lin et al. (2014b) | 200k | 250k | ✓ | ✓ | | | | |
| CrowdPose Li et al. (2019) | 20k | 80k | | ✓ | | | | |
| OCHuman Zhang et al. (2019) | 4.7k | 8k | ✓ | ✓ | | | | |
| ExlPose Lee et al. (2023) | 2.5k | 14k | | ✓ | | | | |
| Human-Art Ju et al. (2023) | 50k | 123k | | ✓ | | | | |
| PoseTrack2018 Andriluka et al. (2018) | 46k | 144k | | ✓ | ✓ | | | |
| PoseTrack21 Doering et al. (2022) | 66k | 177k | | ✓ | ✓ | | | |
| InclusiveVidPose (Ours) | 327k | 309k | ✓ | ✓ | ✓ | ✓ | ✓ | ✓ |

InclusiveVidPose not only demonstrates the limitations of standard HPE models when applied to people with limb deficiencies but also provides a rigorous benchmark for developing more inclusive HPE algorithms. We evaluate six methods on InclusiveVidPose and find that they produce unreliable predictions for missing or prosthetic limbs. Since existing HPE metrics (*i.e.*, OKS (Lin et al., 2014a)) do not account for a model's ability to recognize missing limbs, we propose a new metric, namely Limb-specific Confidence Consistency (LiCC), to quantify this effect. LiCC measures how well the predictions adhere to anatomical exclusion rules. Our results highlight the need for techniques that yield well-calibrated pose estimations and generalize across all body types.

## 2 RELATED WORK

### 2.1 HUMAN POSE DATASETS

High-quality pose datasets are foundational to progress in human pose estimation (HPE), with numerous recent releases offering varied formats and focusing on scale, activity diversity, and visual complexity. Among 2D image-based datasets, MSCOCO (Lin et al., 2014b) is widely used for its large-scale annotations, while MPII Human Pose (Andriluka et al., 2014) emphasizes action diversity. CrowdPose (Li et al., 2019) and OCHuman (Zhang et al., 2019) focus crowded scenes. Domain-specific datasets include HumanArt (Ju et al., 2023) for natural-artificial scene bridging, LSP (Johnson & Everingham, 2011) for sports, and ExLPose (Lee et al., 2023) for extreme lighting conditions. Video-based datasets like PoseTrack2018 (Andriluka et al., 2018) and PoseTrack21 (Doering et al., 2022) provide per-frame body keypoint annotations. Additionally, 3D datasets such as Human3.6M (Ionescu et al., 2014), GoPose (Ren et al., 2022), and FreeMan (Wang et al., 2024) support depth-aware modeling for 3D pose estimation. While these datasets address various challenges in HPE, they consistently operate under the assumption of anatomically intact individuals, overlooking individuals with limb deficiencies. This gap raises concerns around fairness, inclusivity, and limits the generalizability of existing HPE models in real-world diverse populations.

Compared to the datasets in Table 1, InclusiveVidPose uniquely centers on individuals with limb deficiencies, providing a large-scale, video-based resource. The participants cover a wide range of anatomical variations and prosthetic types, including differences in the shape of residual limbs and the appearance of prostheses. InclusiveVidPose covers a wide range of limb difference types, including acquired amputations and congenital conditions. Furthermore, we leverage temporal continuity to improve residual limb keypoint annotation, addressing ambiguity common in image-based datasets. We provide rich annotations including 2D poses, segmentation masks, bounding boxes, and tracking IDs, supporting tasks such as pose estimation, pose tracking, and motion analysis.

### 2.2 HUMAN POSE ESTIMATION MODELS

With increasing demand for real-world applications, human pose estimation has rapidly advanced, with diverse models emerging to optimize accuracy, efficiency, and robustness. Accuracy-oriented models include ViTPose (Xu et al., 2022), which leverages Vision Transformers, AlphaPose (Fang et al., 2022), known for robust multi-task performance, OpenPose (Cao et al., 2019), a pioneering bottom-up approach, DWPose(Yang et al., 2023) which employs a two-stage distillation strategy with depth prediction. Lightweight models, such as YOLOPose (Maji et al., 2022) and RTMPose (He

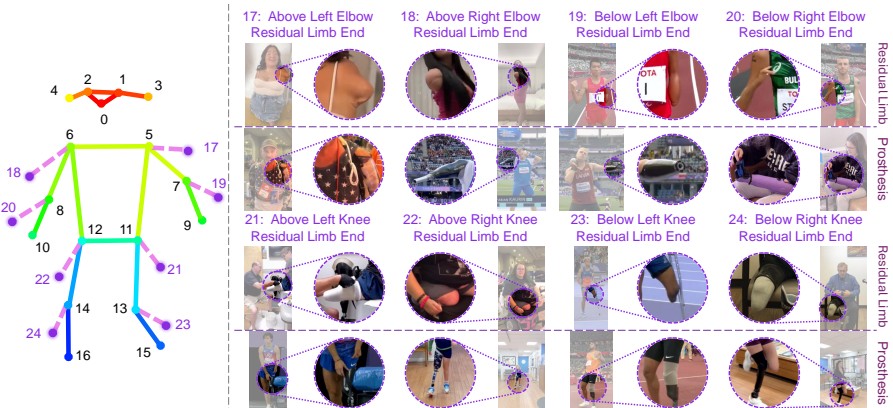

Figure 3: **Demonstration of Our Keypoint Schema.** Our extended pose definition is built on the MS COCO keypoint schema by adding eight residual limb end keypoints (17 through 24). The left panel shows the full skeleton: original COCO keypoints are connected by solid lines, and the eight new residual limb end keypoints are marked as purple circles. The right panel presents, for each residual limb end keypoint, a cropped view of a subject with the corresponding amputation and a matching view of the same subject wearing a prosthetic device.

et al., 2024), are designed for real-time deployment, prioritizing efficiency and fast inference. In addition, DEKR (Geng et al., 2021) introduces decoupled keypoint regression to improve efficiency in multi-person pose estimation. SAPIENs (Khirodkar et al., 2024) have demonstrated the benefits of synthetic data integration, improving generalization across diverse real-world contexts.

## 2.3 HUMAN POSE ESTIMATION FOR HEALTHCARE

Millions globally live with limb deficiencies, facing physical and psychological challenges such as limited mobility, reduced independence, and mental health impacts (Martin, 2013; Day et al., 2019). Advances in assistive technologies including prosthetics (Farina et al., 2023; Andersen et al., 2005; Olsen et al., 2021), motion tracking and analysis tools (Zhou & Hu, 2008; Parks et al., 2019), have improved quality of life. Meanwhile, pose estimation has gained traction in healthcare for applications like rehabilitation and mobility assessment, such as (Alruwaili et al., 2023; 2024; Zhang et al., 2024). Efforts like WheelPose (Huang et al., 2024), which uses synthetic data, and WheelPoser (Li et al., 2024), which leverages IMU signals, have aimed to support pose estimation for marginalized groups. ProGait (Yin et al., 2025) is a 2025 published dataset focus on transfemoral prosthesis users and explicitly focused on lower-limb gait. Compared with these datasets, ours covers more limb deficiency types beyond transfemoral gait, includes whole-body and residual limb keypoints with personalized schema, and adds frame-level prosthesis status, segmentation, boxes, and tracking on videos.

## 3 INCLUSIVEVIDPOSE: FIRST VIDEO POSE DATASET FOR LIMB DEFICIENCIES

### 3.1 EMERGING CHALLENGES OF HPE FOR INDIVIDUALS WITH LIMB DEFICIENCIES

Human pose estimation for individuals with limb deficiencies introduces fundamental challenges that are not addressed by existing models and datasets. Existing keypoint-based systems rely on the presence of anatomical keypoints (*e.g.*, wrists, ankles) that may be absent, altered, or anatomically nonstandard in this population, often resulting in invalid or anatomically implausible pose predictions. Prosthetic limbs further complicate estimation due to their wide variability in geometry, material, articulation, and movement patterns, which are unseen in prior training data. As illustrated in Figure 2, state-of-the-art models frequently fail when encountering missing joints or prosthetic replacements.

In addition, annotation schemes designed for able-bodied subjects do not generalize to limb deficiencies. Missing joints must be modeled explicitly, and residual limbs require alternative keypoints. A single fixed skeleton cannot capture this diversity, motivating flexible per-individual schemas and a dedicated video-based dataset.

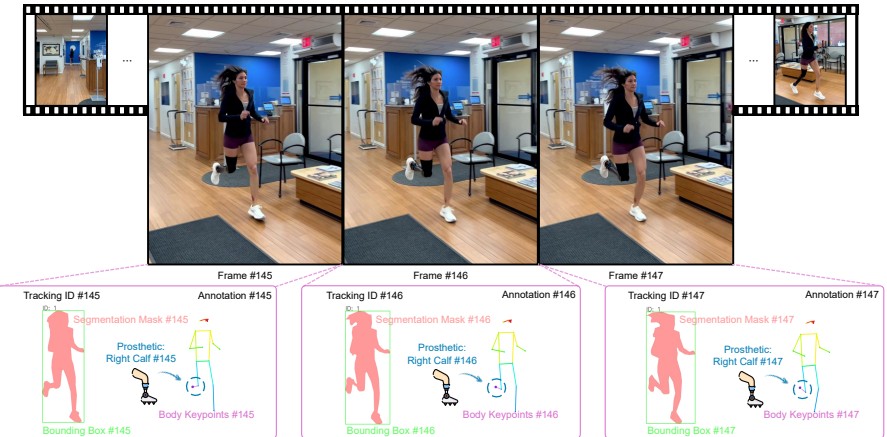

Figure 4: **Overview of frame-by-frame annotations.** Consecutive video frames are shown on the top. For each individual with limb deficiency on every frame, we provide a pixel-level segmentation mask, a bounding box, a set of body keypoints, a prosthesis status label, and a tracking ID.

## 3.2 DATA COLLECTION

**Why do we build our dataset from videos instead of curated images?** In a single image, an occluded limb can appear identical to a missing one, creating ambiguity in annotation. Video sequences provide temporal continuity and changing viewpoints. By observing motion or shifts in perspective, annotators can distinguish a limb that is simply hidden from a limb that is truly absent. This capability leads to precise labeling of residual-limb end keypoints and reduces errors in pose estimation for individuals with limb deficiencies.

**Data sourcing.** We collect raw footage from archival materials from the International Paralympic Committee (IPC) and YouTube videos. For IPC materials, we have gathered the permission to store and distribute the content for academic research. Our data collected from YouTube meet the standard for the fair use (U.S.) and fair dealing for research and study (Australia). Furthermore, the project was approved by the institutional Human Research Ethics Committee in September 2024, with approval valid until September 2029. All frames are annotated at the highest available resolution. We preserve associated metadata for traceability, including upload date, user handle, and caption. We plan ongoing maintenance, including periodic refreshes and corrections to annotations.

**Data filtering and curation.** We manually review all collected data and extract segments that center on individuals with limb deficiencies. We remove clips with severe motion blur or heavy occlusion, such as splashing water, whenever annotators could not agree on keypoint placement. This process yields 313 high-quality videos that span diverse actions, backgrounds and lighting conditions. These curated video segments form the core of our InclusiveVidPose dataset.

## 3.3 HUMAN BODY KEYPOINT SCHEMA FOR INDIVIDUALS WITH LIMB DEFICIENCIES

**Necessity of specialized human body keypoint schema.** Current pose estimation task uses fixed keypoint sets that fail to capture anatomical variation in individuals with limb deficiencies. Models trained on existing datasets often incorrectly predict complete limbs even when these structures are absent or replaced by prostheses. This mismatch yields invalid outputs and reduces reliability. The core issue is the lack of a schema that adapts to residual anatomy. Without keypoints at residual-limb endpoints, methods cannot represent true motion. In light of this, adopting a specialized set of keypoints would address this gap by aligning keypoint definitions with individual anatomy. This alignment is essential to deliver accurate, relevant, and robust pose estimates for individuals with limb deficiencies.

**Design of an extended keypoint schema.** As seen in Figure 3, we adopt a keypoint schema that builds on the COCO format's 17 keypoints by adding 8 residual-limb points. These eight keypoints lie at the anatomical endpoints of residual limbs (left/right above and below the elbow, left/right above and below the knee) and explicitly exclude any prosthetic or assistive device. In this way, the schema focuses on human anatomy rather than device geometry: residual-limb endpoints are

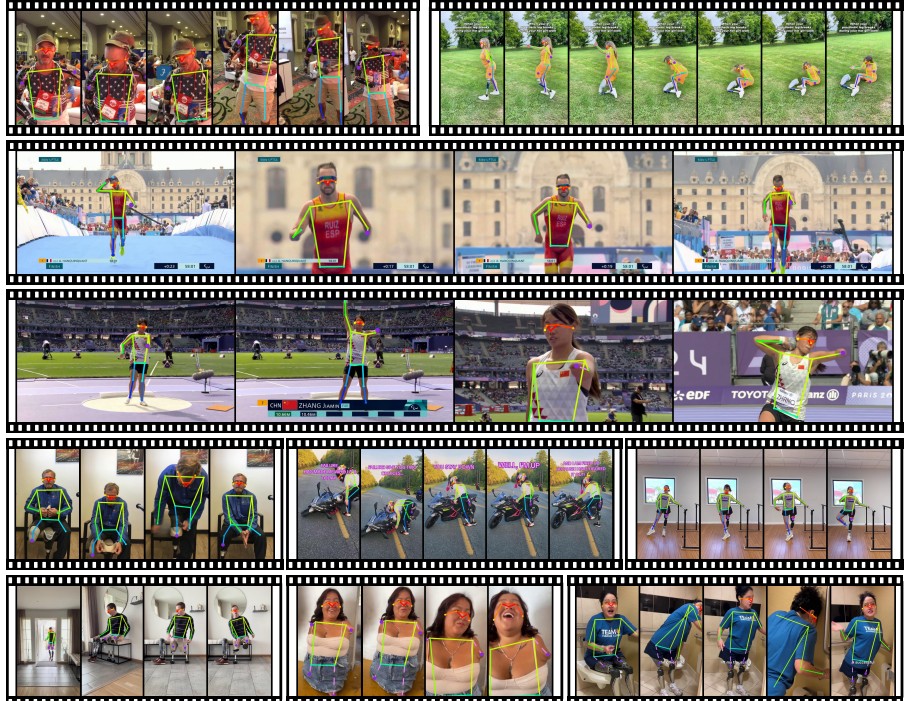

Figure 5: **Overview of our InclusiveVidPose dataset.** Sample frames show individuals with a range of limb-difference anatomies, including upper-body impairments, lower-body impairments, single residual limbs, and multiple residual limbs. Meanwhile, our dataset covers indoor and outdoor environments and a wide variety of activities such as rehabilitation training, sports competition, and everyday interactions. All residual limb points and corresponding skeletons are highlighted in purple.

annotated as stable anatomical landmarks, while prosthesis shape and contact with the environment are represented through pixel-level segmentation masks and per-limb prosthesis status. Standard pose-estimation models lack a mechanism to represent missing or truncated anatomy, so they often try to localize non-existent joints or collapse output when a limb is absent. By incorporating endpoints that correspond to the actual residual anatomy, our 25-keypoint protocol gives models clear, semantically meaningful targets that distinguish between intact and residual structures.

## 3.4 DATA ANNOTATION PROCEDURE

We annotate every individual with five label types: pixel-level segmentation masks, bounding boxes, a persistent tracking ID, limb-deficient body keypoints, and prosthesis limbs, as illustrated in Figure 4.

**Annotation team training.** We recruit 12 annotators through a professional data labeling service. All had prior experience with human body keypoint annotation. Before beginning work, each annotator completed a focused training session led by an internationally accredited para-athletics classifier. This preparation ensured that every team member understood the anatomical variation found in individuals with limb deficiencies and the requirements of our extended keypoint schema.

**Bounding box, segmentation mask and tracking ID.** We use the open-source platform X-AnyLabeling (Wang, 2023) together with Segment Anything 2 promptable segmentation to generate initial masks. For each individual, annotators first merge all segmentation masks sharing the same tracking ID in a given frame and compute the tightest enclosing bounding box around that merged mask. They then refine a pixel-accurate segmentation mask and confirm the persistent tracking ID. SAM2's ability to generalize zero-shot to unseen residual-limb shapes cuts mask-drawing time by over 50%. To maintain high quality, we enforce an 80% accuracy threshold: after one annotator finishes a batch, a second annotator cross-checks it. Any batch with more than twenty percent of masks showing clear errors is returned for revision.

**Human body keypoint annotation process.** To capture each subject's unique anatomy, two trained annotators and one accredited para-athletics classifier review all 398 individuals' disability profiles and agree on a personalized 25-keypoint schema. Each schema is represented by a 25-element

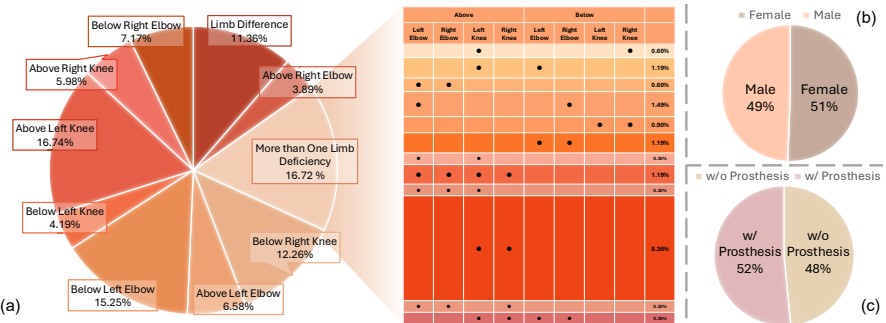

Figure 6: **Overview of Participant Distribution by Deficiency Type, Gender, and Prosthesis.** (a) Distribution of limb-deficiency types, with single-site deficiencies and multi-site cases. (b) Gender Distribution. (c) Distribution of individuals with and without prosthetic limbs.

presence mask that indicates which of the 25 keypoints apply to that individual. Annotators then label every frame according to that schema, marking each keypoint's image coordinates. We leverage temporal cues to enforce cross-frame consistency and smooth annotations over frames. In addition, for every frame, we record a prosthesis status label that specifies which limb or limbs are fitted with a prosthesis. We follow the COCO visibility convention: any keypoint that is within the frame but occluded receives a visibility flag of 1. We require an 80% point-level agreement rate between annotators on 5% of sampled data, where "agreement" means no visually apparent errors in keypoint placement. If more than 20% of sampled points in a batch fall below this threshold, the original annotator must correct the entire batch. This structured, expert-guided workflow ensures unambiguous placement of residual-limb keypoints and delivers highly reliable pose data.

## 3.5 DATASET STATISTICS

Our dataset comprises 313 video sequences, capturing the motion of 398 unique individuals. In total, we annotate 327,235 frames and produce 308,533 pose estimates. As shown in Figure 5, our dataset encompasses a broad spectrum of limb deficiencies. As illustrated in Figure 6, (a) the most prevalent single-site deficiencies are located above the left knee (16.74%), below the left elbow (15.25%), and below the right knee (12.26%); Multi-site limb differences account for 16.72% of the population, highlighting the complexity of representation beyond isolated cases and the necessity of modeling diverse impairment patterns for robust performance. The co-occurrence matrix further demonstrates that upper- and lower-limb deficiencies frequently overlap across different body sides, reflecting realistic clinical patterns. The gender breakdown in (b) is nearly balanced (51% female, 49% male), ensuring that learned pose models do not inherit a strong gender bias. Finally, the prosthetic-usage chart in (c) shows an even split between subjects with and without prostheses, indicating that the dataset equally represents natural residua and prosthetic-assisted movements. Together, these statistics demonstrate that the dataset provides a representative and diverse sample of limb-deficiency conditions, gender, and assistive-device usage, which is critical for developing generalizable human-pose estimation algorithms.

**Data use and license.** The annotations and all website content for the InclusiveVidPose dataset are © 2025 InclusiveVidPose Consortium and are released under the Creative Commons BY-NC-SA 4.0 license ©. We will continue to maintain the dataset and release versioned updates with changelogs.

## 4 EXPERIMENTS

### 4.1 EXPERIMENT DETAILS

We benchmark our InclusiveVidPose dataset on a single-frame pose estimation task. Single-frame evaluation highlights exactly where models struggle with residual-limb endpoints, guiding targeted improvements in keypoint localization and confidence calibration. As the first pose-estimation dataset dedicated to individuals with limb deficiencies, this image-based evaluation establishes clear baselines for residual-limb detection and ensures fair comparison across models by using identical single-frame inputs and metrics, while also informing future extensions to multi-frame pose estimation.

Table 2: **Main experimental results on pose estimation algorithms.** We evaluate models including Swin-based top-down heatmap networks (Liu et al., 2021a), ViTPose (Xu et al., 2022), and RTMPose (He et al., 2024), as well as the bottom-up model DEKR (Geng et al., 2021), the detector-based single-stage model YOLOX-Pose (Maji et al., 2022), and the video-based ViPNAS (Xu et al., 2021). "InclusiveVidPose → InclusiveVidPose" reports training on our training set and evaluation on our validation and test splits. "COCO → InclusiveVidPose" reports training only on COCO and evaluation on our InclusiveVidPose validation and test splits. "InclusiveVidPose + COCO → InclusiveVidPose/COCO" reports training on both datasets, validating on our validation split and testing on our test split and COCO validation. "COCO → COCO" reports COCO training and COCO validation performance (from MMPose).

| Method | Backbone | Input Size | InclusiveVidPose → InclusiveVidPose | | | | | | | COCO | | InclusiveVidPose + COCO → InclusiveVidPose | | | | | | | → COCO | | COCO → COCO | |
|---|---|---|---|---|---|---|---|---|---|---|---|---|---|---|---|---|---|---|---|---|---|---|
| | | | AP | AP50 | AP75 | AR | AR50 | AR75 | LiCC | AP | AR | AP | AP50 | AP75 | AR | AR50 | AR75 | LiCC | AP | AR | AP | AR |
| YoloxPose | YoloxPose-T | 416x416 | 60.1 | 75.3 | 63.9 | 73.3 | 88.0 | 76.0 | 70.7 | 74.0 | 91.1 | 70.3 | 79.8 | 74.1 | 84.9 | 94.5 | 88.0 | 78.7 | 41.7 | 46.9 | 52.6 | 57.1 |
| | YoloxPose-S | 640x640 | 65.4 | 80.2 | 69.9 | 77.5 | 90.5 | 80.7 | 72.7 | 78.4 | 93.3 | 73.6 | 82.3 | 77.0 | 87.4 | 95.0 | 89.8 | **78.0** | 49.1 | 54.2 | 64.1 | 68.2 |
| | YoloxPose-M | 640x640 | 65.7 | 80.5 | 70.0 | 76.8 | 89.1 | 80.4 | 69.9 | 79.5 | 94.4 | 74.0 | 83.7 | 77.3 | 86.5 | 95.6 | 90.6 | 72.0 | 59.4 | 74.5 | 69.5 | 89.9 |
| | YoloxPose-L | 640x640 | 65.9 | 80.3 | 70.4 | 76.2 | 88.5 | 79.9 | 67.7 | 80.0 | 94.9 | 74.3 | 84.1 | 77.5 | 85.3 | 94.7 | 89.1 | 69.5 | 64.9 | 80.5 | 71.2 | 90.1 |
| DEKR | HRNet-w32 | 512x512 | 77.7 | 83.5 | 79.5 | 83.2 | 89.9 | 84.8 | 55.2 | 55.8 | 93.2 | 73.6 | 78.2 | 74.9 | 89.3 | 96.2 | 90.5 | 62.1 | 59.5 | 66.3 | 68.6 | 73.5 |
| | HRNet-w48 | 640x640 | 75.2 | 81.1 | 77.2 | 85.1 | **93.7** | 86.7 | 67.3 | 53.4 | 93.4 | 71.1 | 75.3 | 72.1 | **90.6** | **97.2** | **91.7** | 55.2 | 62.9 | 69.2 | 71.4 | 76.2 |
| ViPNAS | MobileNetV3 | 256x192 | 78.6 | 87.6 | 80.3 | 80.3 | 88.6 | 81.5 | 53.8 | 70.2 | 72.9 | 81.5 | 88.6 | 83.3 | 83.2 | 89.8 | 84.3 | 55.0 | 68.5 | 72.2 | 69.5 | 75.5 |
| Swin Transformer | Swin-T | 256x192 | 81.2 | 89.6 | 83.3 | 82.6 | 90.0 | 84.5 | 68.8 | 72.6 | 75.8 | 82.0 | 88.6 | 84.4 | 83.5 | 89.9 | 85.3 | 68.9 | 68.3 | 71.7 | 72.4 | 78.2 |
| | Swin-B | 256x192 | 78.8 | 87.5 | 81.1 | 80.6 | 88.4 | 82.6 | 67.8 | 77.3 | 80.2 | 80.7 | 88.5 | 83.1 | 82.5 | 89.5 | 84.2 | 69.3 | 68.6 | 72.5 | 73.7 | 79.4 |
| | Swin-B | 384x288 | 81.8 | 88.6 | 83.4 | 83.2 | 89.8 | 84.5 | 69.3 | 78.8 | 81.6 | 82.7 | 89.4 | 84.3 | 84.1 | 90.4 | 85.4 | 74.6 | 70.6 | 74.7 | 75.9 | 81.1 |
| | Swin-L | 256x192 | 79.5 | 88.4 | 81.1 | 81.1 | 89.5 | 82.3 | 68.7 | 77.1 | 79.9 | 80.0 | 88.4 | 82.1 | 81.9 | 89.2 | 83.9 | 68.2 | 65.2 | 69.3 | 74.3 | 79.8 |
| | Swin-L | 384x288 | 80.7 | 88.6 | 82.5 | 82.0 | 89.4 | 83.6 | 72.1 | 78.4 | 81.4 | 81.5 | 88.6 | 83.5 | 82.8 | 89.2 | 84.5 | 72.7 | 66.0 | 70.7 | 76.3 | 81.4 |
| RTMPose | RTMPose-T | 256x192 | 75.2 | 83.1 | 76.2 | 76.4 | 84.9 | 77.9 | 65.3 | 66.5 | 68.8 | 80.2 | 89.7 | 82.5 | 81.6 | 90.5 | 83.7 | 73.1 | 67.6 | 70.9 | 68.2 | 73.6 |
| | RTMPose-S | 256x192 | 79.3 | 84.5 | 77.3 | 80.7 | 86.4 | 78.7 | 74.1 | 71.8 | 74.0 | 83.6 | 90.8 | 85.6 | 84.8 | 91.2 | 96.3 | 71.4 | 72.2 | 75.1 | 71.6 | 76.8 |
| | RTMPose-M | 256x192 | 82.2 | 88.8 | 83.6 | 83.3 | 89.9 | 84.3 | 69.5 | 75.2 | 77.7 | 82.5 | 89.6 | 83.4 | 83.7 | 90.2 | 84.7 | 60.8 | 71.9 | 74.7 | 74.6 | 79.5 |
| | RTMPose-L | 256x192 | 82.2 | 89.7 | 83.5 | 83.2 | 90.1 | 84.1 | 69.1 | 76.2 | 78.7 | 81.9 | 89.6 | 83.2 | 83.5 | 90.1 | 84.6 | 60.8 | 72.6 | 75.5 | 75.8 | 80.6 |
| ViTPose | ViT-S | 256x192 | 82.5 | 89.7 | 84.4 | 84.1 | 90.7 | 85.7 | 67.1 | 72.5 | 75.2 | 82.1 | 88.4 | 84.2 | 93.7 | 90.0 | 85.0 | 70.6 | 70.2 | 73.7 | 73.9 | 79.2 |
| | ViT-B | 256x192 | 82.5 | 89.6 | 84.2 | 84.1 | 90.2 | 85.6 | 70.3 | 77.4 | 80.0 | 83.3 | 89.5 | 85.4 | 84.9 | 90.4 | 86.5 | 72.0 | 74.3 | 77.6 | 75.7 | 81.0 |
| | ViT-L | 256x192 | 85.5 | 90.7 | 86.5 | 86.8 | 91.6 | 87.3 | **73.8** | 79.9 | 82.5 | 86.0 | 91.6 | 87.4 | 87.4 | 92.1 | 88.4 | 75.1 | 78.4 | 81.2 | 78.2 | 83.4 |
| | ViT-H | 256x192 | **86.3** | **90.8** | **87.5** | **87.6** | 91.8 | **88.7** | 73.6 | 81.5 | 83.8 | **86.5** | **91.7** | **87.5** | 87.9 | 92.4 | 88.9 | 74.0 | 79.1 | 82.1 | 78.8 | 83.9 |

Beyond this image-based setting, we further build a video-based benchmark by adapting the PoseTrack evaluation protocol to InclusiveVidPose. We evaluate two representative multi-frame pose estimators, DCPose (Liu et al., 2021b) and DSTA (He & Yang, 2024), and report PoseTrack-style AP for standard joints and for our residual-limb groups. This video-based evaluation tests whether temporal aggregation helps track residual-limb endpoints across time and offers the first reference results for video pose estimation on individuals with limb deficiencies.

Specifically, we sample one frame every 60 frames from each video. We then split the dataset at the video level into training, validation and test splits in a ratio of 7 : 1 : 2. All sampled frames from the same video are kept together in one split. Therefore, no individual appears in more than one split to prevent any data leakage. Furthermore, we believe that a good pose estimator must serve all users rather than a single group. In light of this, we measure performance on both able-bodied and limb-difference anatomies. We train and evaluate models on COCO dataset alongside InclusiveVidPose under the same settings. This dual-dataset approach provides a fair comparison of how different algorithms handle standard and disability-aware keypoint estimation. By benchmarking across both COCO and InclusiveVidPose, we ensure our evaluations drive improvements that benefit everyone, from able-bodied individuals to those with limb deficiencies. All models in Table 2 are initialized from their official COCO-pretrained weights.

## 4.2 EVALUATION METRICS

**COCO metric** are used to evaluate overall pose estimation performance. We report average precision (AP) and average recall (AR) at OKS thresholds 0.50 and 0.75, *i.e.*, AP, $AP^{50}$, $AP^{75}$, AR, $AR^{50}$, and $AR^{75}$. These metrics follow the COCO protocol and measure keypoint localization accuracy.

**PoseTrack metrics** are used to evaluate video-based pose estimation on InclusiveVidPose. Following the PoseTrack protocol, we compute keypoint average precision (AP) over all frames and report AP for each joint category as well as the mean AP over all keypoints. On our benchmark, we further report AP for the four residual-limb groups (ArmUp, ArmLow, LegUp, LegLow), which aggregate our residual-limb endpoints and quantify localization performance on disability-related regions.

Table 3: **PoseTrack-style keypoint AP on InclusiveVidPose.** We report AP (%) of DCPose (Liu et al., 2021b) and DSTA (He & Yang, 2024) on standard joints and residual-limb groups (ArmUp, ArmLow, LegUp, LegLow), together with the mean over all keypoints.

| | Head | Shoulder | Elbow | Wrist | Hip | Knee | Ankle | ArmUp | ArmLow | LegUp | LegLow | Mean |
|---|---|---|---|---|---|---|---|---|---|---|---|---|
| DCPose (Liu et al., 2021b) | 28.1 | 72.0 | 69.7 | 79.3 | 73.2 | 72.4 | 72.9 | 1.6 | 0.2 | 12.2 | 16.0 | 43.2 |
| DSTA (He & Yang, 2024) | 28.9 | 72.2 | 70.4 | 81.9 | 72.7 | 71.9 | 72.0 | 0.6 | 0.0 | 14.3 | 17.3 | 43.7 |

**Limb-specific Confidence Consistency (LiCC)** is introduced to measure whether a pose-estimation model can correctly distinguish intact limbs from residual or missing limbs. This capability is critical for assessing the inclusiveness of pose-estimation systems on data that includes individuals with limb deficiencies. Let $V$ be the set of all ground-truth keypoints with visibility $v \geq 1$, and for each keypoint $i \in V$, denote by $M(i)$ the set of mutually exclusive keypoints. For example, if the residual wrist keypoint is visible, then both the residual elbow keypoint and the normal wrist landmark cannot be present. Denote by $s_i$ the predicted confidence for keypoint $i$, and by $\max_{j \in M(i)} s_j$ the highest confidence among its mutually exclusive partners. LiCC is defined as the average fraction of keypoints whose confidence exceeds that of any exclusive keypoint:

$$\text{LiCC} = \frac{1}{|V|} \sum_{i \in V} \mathbf{1}\left(s_i > \max_{j \in M(i)} s_j\right), \tag{1}$$

where $\mathbf{1}(\cdot)$ is the indicator function. A higher LiCC indicates stronger consistency: visible keypoints are assigned higher confidence than any impossible alternatives. Pseudocode for computing LiCC is provided in the supplementary materials.

### 4.3 BENCHMARK RESULTS

**Single-frame pose estimation.** As shown in Table 2, when we train on InclusiveVidPose and evaluate on InclusiveVidPose, large backbones like HRNet-w48, Swin-L, and ViTPose (ViT-H/L) get higher AP scores on visible points. However, LiCC is still low for most methods, about 60% in many cases. This means many residual keypoints are predicted wrongly. Existing models assume that all keypoints exist, so they often predict high confidence for impossible points. DEKR and ViPNAS have good COCO AP but low LiCC, so they do not tell intact joints from residual endpoints well. In contrast, YOLOX-Pose, which uses confidence learning, gets higher LiCC scores because it separates intact and residual cases better.

To make the distribution shift between COCO and InclusiveVidPose explicit, Table 2 also reports a *COCO→InclusiveVidPose* setting. We train all models only on COCO and evaluate them on our validation and test splits, using a 17-keypoint version of InclusiveVidPose that keeps only the COCO joints and ignores the 8 residual endpoints. Compared with *COCO→COCO* under the same 17-keypoint label space, many methods show lower AP or AR and their ranking changes, which indicates that InclusiveVidPose presents a different and more challenging test distribution. At the same time, strong models such as ViTPose-H and YOLOX-Pose-L still reach high AP and AR (e.g., 81.5 AP / 83.8 AR and 80.0 AP / 94.9 AR in the *COCO→InclusiveVidPose* setting), which shows that COCO-trained models can transfer reasonably well to our 17 keypoint annotations. This behaviour is more consistent with a real distribution shift between COCO and InclusiveVidPose, combined with different generalization strengths across architectures, than with inconsistent keypoint annotations in our dataset.

Furthermore, we add COCO to training. This helps large backbones on InclusiveVidPose, but it hurts small models like YOLOX-Pose-T and RTMPose-T on people with limb differences. A fair comparison under a fixed label space therefore compares *COCO→InclusiveVidPose* to *COCO→COCO* to measure dataset shift, and compares *InclusiveVidPose→InclusiveVidPose* and *InclusiveVidPose+COCO→InclusiveVidPose* to understand what in-domain training can achieve on the same 17 standard keypoints. *COCO→COCO* also drops compared with a COCO-only baseline that predicts only 17 keypoints, because the head must now predict eight extra residual-limb keypoints. The output space is larger, and the capacity moves away from the original 17 points. Additional results on the endpoints are in Appendix §D, including results on 8 endpoints and 17 keypoints.

Overall, LiCC is low across most methods. On InclusiveVidPose, many models are around 60% LiCC. In many frames, the model gives higher confidence to an anatomically impossible point than

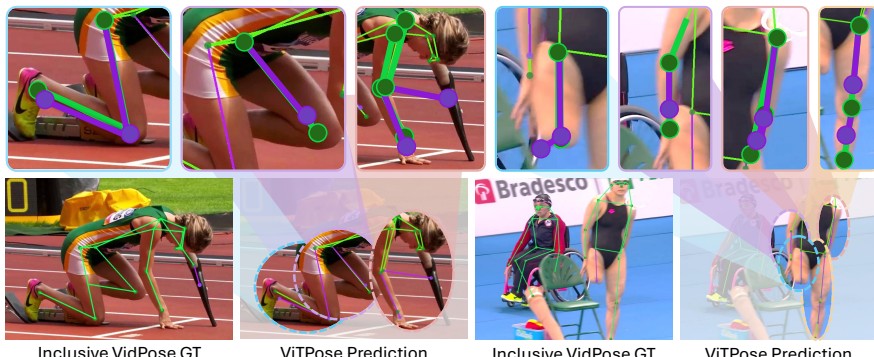

Figure 7: **Case study of ViTPose (ViT-B) predictions trained on InclusiveVidPose.** We highlight incorrect keypoints and skeleton segments in bold, demonstrating the challenges introduced InclusiveVidPose. Residual-limb endpoints appear in purple, and all other keypoints appear in green.

to the true residual point. This shows that current pose estimators do not handle missing or altered limbs. Our dataset and the LiCC metric make this problem clear and give a target for better methods.

**Case study.** As demonstrated in Figure 7, these results show that existing methods fail to generalize effectively to individuals with limb deficiencies. More than half of the visible keypoint instances exhibit conflicting predictions, underscoring the importance of our dataset for developing and evaluating algorithms that can handle anatomies with missing or altered limbs. These errors occur across activities and views, showing that InclusiveVidPose is challenging and calling for new, more inclusive models that can reason about absent joints and residual endpoints.

**Multi-frame pose estimation.** Table 3 reports PoseTrack-style keypoint AP on InclusiveVidPose for two representative video pose estimators, DCPose and DSTA. Both models keep relatively high AP on standard joints: shoulders, elbows, wrists, hips, knees, and ankles all stay around the low to mid 70s, and DSTA gives slightly higher scores than DCPose on most of these joints, especially the wrist. Head AP is much lower for both methods, around 28 AP, which is consistent with frequent occlusion, cropping, and motion blur in our videos.

In contrast, performance on the four residual-limb groups is very poor. For ArmUp and ArmLow, AP is almost zero for both methods, and even for LegUp and LegLow, AP remains in the low to mid teens. The mean AP over all keypoints only increases from 43.2 for DCPose to 43.7 for DSTA, which shows that small gains on intact joints and lower limbs do not close the large gap on residual-limb regions. These results indicate that state-of-the-art video pose estimators, which are mainly trained and tuned on intact-body datasets, do not transfer to our residual-limb keypoints, even after we aggregate eight endpoints into four coarse groups. InclusiveVidPose therefore defines a new evaluation regime where models need to reason about diverse amputation levels, prosthetic shapes, and asymmetric limb geometry. By making this failure mode measurable in a realistic video benchmark, our dataset supports both scientific progress toward more general human motion understanding and practical advances in inclusive analysis for rehabilitation, sports, and everyday activities of people with limb differences.

## 5 Conclusion

In summary, we introduce InclusiveVidPose, the first large-scale video-based human pose estimation dataset focused on individuals with limb deficiencies, addressing a critical gap left by existing benchmarks that only represent able-bodied anatomies. By capturing diverse anatomical variations with precise annotations, including novel residual-limb end keypoints and leveraging video context to disambiguate occlusions from absent limbs, our dataset enables more accurate modeling of this underserved population. Our evaluation of state-of-the-art methods reveals their limitations in handling missing or prosthetic limbs, motivating our proposed Limb-specific Confidence Consistency (LiCC) metric to better assess confidence calibration. We believe InclusiveVidPose and LiCC will drive the development of more inclusive and robust pose estimation algorithms, benefiting applications in rehabilitation, health assessment, and assistive technologies.

ETHICS STATEMENT

This work uses publicly available videos from YouTube and archival materials from the International Paralympic Committee (IPC). No new data are collected from human subjects, and no interventions are conducted. Use of YouTube content follows fair use in the United States and fair dealing for research and study in Australia (see §3.2). We obtained guidance from the institutional copyright officer and approval from the institutional Human Research Ethics Committee in September 2024, valid until September 2029.

We release annotations and documentation. We do not redistribute YouTube videos. For YouTube content, we provide links and provenance only. For IPC materials, we are authorized to store and distribute the content for academic research. We remove content upon substantiated takedown requests. We retain only information already public on the source platforms (e.g., upload date and user handle) for traceability.

This work aims to advance inclusive human pose estimation. By curating representative video data and releasing carefully documented annotations, we enable more reliable benchmarking and model development for individuals with limb deficiencies. The dataset supports fairer evaluation, fosters methodological innovation on non-standard anatomies, and facilitates downstream applications in rehabilitation analytics, accessible sports technology, and safety-critical monitoring. We expect these resources to help the community design models that better reflect human diversity and improve real-world accessibility.

We document known limitations, report disaggregated metrics where applicable (§4). Annotations and website content are released under CC BY–NC–SA 4.0 (see §3.5), and downstream users must follow applicable laws and institutional ethics requirements. The authors declare no conflicts of interest and no sponsorship that influenced the study.

REPRODUCIBILITY STATEMENT

We aim to make all experiments reproducible. Data sourcing and licensing are described in §3.2 and §3.5. Dataset construction, annotation protocol, and quality controls are detailed in §3.4. Dataset statistics and splits are provided in §3.5 and §4.1. The benchmark tasks, metrics, and evaluation protocol appear in §4. We release an anonymized repository with training and evaluation scripts, configuration files, hyperparameters, model variants, seeds, and environment specifications at </> code.

ACKNOWLEDGMENTS

This research is funded in part by ARC-Discovery grant (DP220100800 to XY and DP230101753) and ARC-DECRA grant (DE230100477 to XY). We thank Advance Queensland Industry Research Projects (AQIRP) for their support and guidance. We thank the area chair and anonymous reviewers for constructive comments.

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

## A    STATEMENT ON LLM USAGE

We use a large language model only for language polishing. Its role is limited to grammar, spelling, punctuation, and minor wording edits. It does not contribute to research ideation, study design, data analysis, result interpretation, or substantive writing. All technical content and claims originate from the authors, and every edit receives author review and approval.

## B    DATASET WEBSITE

All information related to the dataset is available on our anonymous project website ⌘ InclusiveVid-Pose.

## C    BOARDER IMPACT

The InclusiveVidPose dataset and accompanying evaluation benchmark aim to address a critical gap in human pose estimation (HPE) research by centering on individuals with limb deficiencies, an underrepresented and clinically important population. We anticipate the following broader impacts:

### C.1    POSITIVE BENEFITS TO COMMUNITY

**Fairness and Exclusivity in AI Development**    By providing a large-scale, anatomically diverse video dataset, we encourage the community to move beyond able-bodied assumptions. This can catalyze new model architectures and training paradigms that generalize across a wider spectrum of human bodies, helping to reduce algorithmic biases in downstream applications (*e.g.*, surveillance and sports analytics).

**Assistive Technologies and Rehabilitation**    Models trained and validated on InclusiveVidPose can be integrated into physical-therapy monitoring tools, prosthetic calibration systems, and home-based exercise feedback platforms for individuals with limb deficiencies. This integration enhances the independence and improves the quality of life for users with limb loss or congenital differences.

**Research Resource**    The dataset includes rich annotations, *i.e.*, keypoints, segmentation masks, bounding boxes, and tracking IDs. These annotations serve as a valuable resource for developing novel pose estimation and biomechanics algorithms and for teaching best practices in inclusive model evaluation.

We hope our work draws greater attention to individuals with disabilities and empowers them to benefit from advancements in technology.

### C.2    POTENTIAL RISKS

**Privacy and Misuse**    Although we only share YouTube URLs and anonymized keypoint data, improper downloading or re-identification attempts could compromise personal privacy. To mitigate this, we require strict anonymization standards and a Data Use Agreement (DUA) that forbids deanonymization, re-identification, and commercial exploitation without consent.

**Unintended Bias in Applications**    Models fine-tuned on limb-deficient data could be misapplied in contexts where missing-joint detection is erroneously interpreted as injury or non-compliance (e.g., automated safety monitoring). We recommend clear documentation of model limitations and ethical training for practitioners deploying these technologies.

## D    FURTHER ANALYSIS OF RESIDUAL ENDPOINTS

In Table 4, we extend the study by reporting ViTPose-H on InclusiveVidPose with 8, 17, and 25 keypoints under two training settings: InclusiveVidPose only and InclusiveVidPose plus COCO, both evaluated on InclusiveVidPose. Under InclusiveVidPose-only training, AP on the 17 standard joints

Table 4: **ViTPose-H results on InclusiveVidPose for 8 residual endpoints and 17 standard joints.** We report performance on three keypoint sets: the 8 residual endpoints, the standard 17 COCO joints, and the extended 25-keypoint schema. The left block (InclusiveVidPose* → InclusiveVidPose) trains ViTPose-H from scratch on InclusiveVidPose without COCO pretraining. The middle block (InclusiveVidPose → InclusiveVidPose) initializes from COCO weights and fine-tunes only on InclusiveVidPose. The right block (InclusiveVidPose+COCO → InclusiveVidPose) jointly trains on COCO and InclusiveVidPose before evaluation on InclusiveVidPose. AP on the 17 standard keypoints is consistently higher than AP on the 8 residual endpoints, which suggests that residual endpoints are harder to localize. Adding COCO, either as initialization or as extra training data, mainly helps the 17-keypoint subset, while accuracy on the 8 residual endpoints changes little.

| | InclusiveVidPose* → InclusiveVidPose | | | | | | InclusiveVidPose → InclusiveVidPose | | | | | | InclusiveVidPose+COCO → InclusiveVidPose | | | | | |
| Keypoints | AP | $AP^{50}$ | $AP^{75}$ | AR | $AR^{50}$ | $AR^{75}$ | AP | $AP^{50}$ | $AP^{75}$ | AR | $AR^{50}$ | $AR^{75}$ | AP | $AP^{50}$ | $AP^{75}$ | AR | $AR^{50}$ | $AR^{75}$ |
|---|---|---|---|---|---|---|---|---|---|---|---|---|---|---|---|---|---|---|
| 8 | 82.4 | 90.1 | 85.4 | 85.2 | 91.6 | 87.2 | 84.2 | 92.2 | 86.3 | 86.7 | 93.1 | 88.0 | 82.2 | 92.0 | 84.7 | 85.7 | 93.1 | 87.3 |
| 17 | 85.2 | 89.6 | 86.3 | 86.6 | 90.9 | 87.4 | 86.7 | 90.8 | 87.5 | 87.9 | 91.8 | 88.6 | 87.0 | 91.6 | 87.4 | 88.3 | 92.3 | 88.8 |
| 25 | 84.8 | 90.6 | 86.3 | 86.3 | 91.1 | 89.4 | 86.3 | 90.8 | 87.5 | 87.6 | 91.8 | 88.7 | 86.5 | 91.6 | 87.5 | 87.9 | 92.4 | 88.9 |

Table 5: **ViTPose-H performance on 17 COCO keypoints by limb-deficiency group.** We report COCO-style AP and AR on InclusiveVidPose using a 17-keypoint subset and group clips into Arm Left, Arm Right, Leg Left, Leg Right residual-limb cases and Intact clips, showing consistent annotation of the shared joints across limb-deficiency types.

| | AP | $AP^{50}$ | $AP^{75}$ | AR | $AR^{50}$ | $AR^{75}$ |
|---|---|---|---|---|---|---|
| Arm Left | 84.7 | 88.4 | 85.9 | 86.6 | 89.9 | 87.4 |
| Arm Right | 82.3 | 86.1 | 82.9 | 86.2 | 89.7 | 86.8 |
| Leg Left | 92.3 | 95.9 | 92.9 | 93.0 | 96.1 | 93.7 |
| Leg Right | 87.5 | 91.9 | 88.6 | 88.7 | 93.0 | 89.1 |
| Intact | 75.7 | 84.1 | 76.4 | 81.5 | 88.1 | 83.0 |

is 86.7, while AP on the 8 residual endpoints is 84.2, which suggests that the standard joints are easier. This is because most subjects have only one or two residual endpoints, their appearance varies across individuals, and occlusion near amputation sites is common. When COCO is added to training, the main gains appear on the 17-joint subset (AP increases to 87.0 and AR to 88.3). The 8-endpoint subset does not improve (AP decreases to 82.2 and AR to 85.7), and the unified 25-keypoint set changes only slightly (AP moves from 86.3 to 86.5). These results suggest that large general datasets mainly strengthen common joints, while the residual endpoints still need targeted data and modeling.

To isolate the effect of COCO pretraining, we add an "InclusiveVidPose* → InclusiveVidPose" setting, where ViTPose-H is trained from scratch on InclusiveVidPose without any COCO initialization. Comparing this baseline with "InclusiveVidPose → InclusiveVidPose" shows that COCO pretraining brings consistent gains across all keypoint sets: AP/AR improve from 82.4/85.2 to 84.2/86.7 on the 8 residual endpoints, from 85.2/86.6 to 86.7/87.9 on the 17 standard joints, and from 84.8/86.3 to 86.3/87.6 on the full 25-keypoint schema. The improvements are similar in magnitude for residual endpoints and standard joints, which suggests that pretraining on a large intact-body dataset provides a useful generic pose prior without harming residual-endpoint localization. Combined with the joint-training results, these trends indicate that COCO mainly acts as a strong initialization that stabilizes common body structure, while closing the remaining gap between standard joints and residual endpoints still requires targeted InclusiveVidPose-style data and modeling.

We further evaluate ViTPose-H on a 17-keypoint version of InclusiveVidPose and group clips into five categories: Arm Left, Arm Right, Leg Left, Leg Right, and Intact. As shown in Table X, ViTPose-H achieves high AP and AR for all four limb-deficiency groups (around 82–85 AP for residual-arm clips and 87–92 AP for residual-leg clips), while the Intact group is noticeably lower (75.7 AP and 81.5 AR). This is expected because, for limb-deficiency clips, keypoints on the missing limb are marked with visibility = 0 and are ignored when COCO metrics are computed, so the evaluation only covers the visible standard joints. In contrast, intact clips include a larger set of visible joints, including more occluded and fast-moving points that are harder to localize. The gap between arm and leg groups suggests that upper-limb joints near the torso remain more challenging than lower-limb

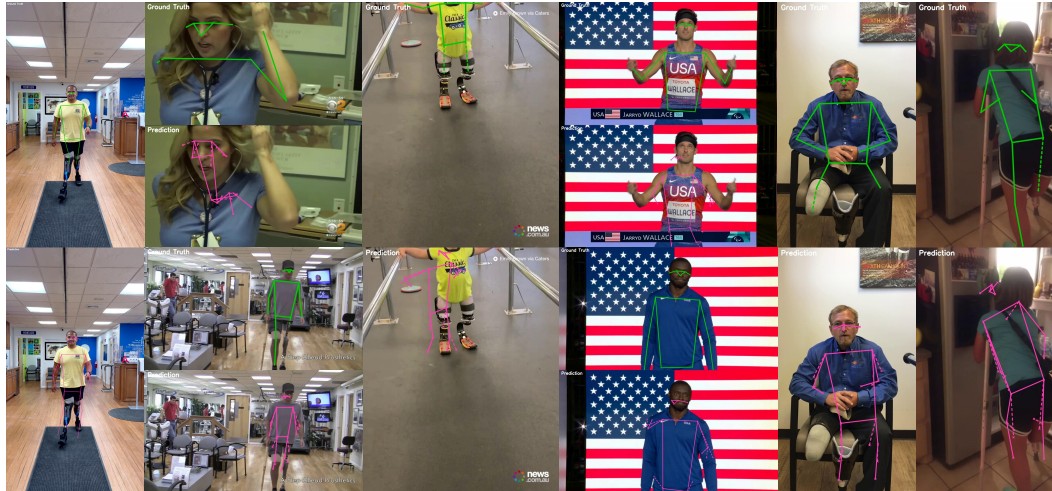

Figure 8: **Failure cases of DSTA on InclusiveVidPose.** Each frame shows two rows: the top row visualizes the ground truth keypoints and the bottom row shows predictions from DSTA. All examples come from the InclusiveVidPose test split. Even when the overall pose looks roughly plausible, DSTA often places residual endpoints and nearby joints at anatomically implausible locations relative to the clearly shortened limbs and prostheses in the ground truth.

joints, but across all groups ViTPose-H maintains strong 17-keypoint performance. These results support that the shared COCO joints are annotated consistently across limb-deficiency types and that the main difficulty lies in detecting and localizing residual endpoints rather than in inconsistent placement of standard keypoints.

## E    FURTHER ANALYSIS OF MULTI-FRAME POSE ESTIMATION

In the main paper, we evaluate both image-based pose estimators and recent multi-frame video models, including DCPose (Liu et al., 2021b) and DSTA (He & Yang, 2024), on InclusiveVidPose. For the video benchmark we follow a PoseTrack style evaluation protocol and report mAP over keypoints. All reported numbers use the same 25 keypoint skeleton, the same training split of InclusiveVidPose.

**Head joints.**    The behavior of head keypoints on InclusiveVidPose differs from PoseTrack style benchmarks. Many InclusiveVidPose sequences center on the torso and residual limbs, for example rehabilitation training or prosthesis demonstrations, so the head is often near the crop boundary or partially out of frame. In contrast, PoseTrack videos usually show full body, upright people with clear heads near the image center Andriluka et al. (2018). In addition, facial keypoints are annotated in fewer instances than most torso and leg joints. Each head joint appears only on the order of one hundred thousand labeled instances in our video annotations, which is roughly about half of the supervision frequency that standard lower body joints receive. As a result, all models see noticeably fewer positive examples for the head than for knees, hips, and ankles during training, and head AP is lower than for better supported joints.

**Residual endpoints and temporal assumptions.**    The larger gap on residual limb endpoints comes mainly from how current video pose models are designed. DCPose and DSTA are built to exploit temporal continuity in videos where every subject has a complete skeleton. They always predict a fixed set of joints for each person in every frame, and they aggregate neighboring frames to smooth short term occlusions, blur, or defocus (Liu et al., 2021b; He & Yang, 2024). On PoseTrack this assumption holds, because there are no limb deficiencies and occlusions are usually temporary. In

InclusiveVidPose, residual limbs are structural rather than temporary. Many participants have only one or two residual endpoints and the rest of the limb is truly absent. When a model that assumes a full limb sees such a sequence, it tends to hallucinate a complete arm or leg and gradually pull the residual endpoint toward a typical wrist or ankle location. Once this happens near the beginning of a clip, the temporal module propagates and stabilizes the wrong configuration across many frames. Residual endpoints then follow a smooth but systematically shifted trajectory and receive low AP under per frame evaluation.

The image baselines instead make independent predictions at each frame. They still struggle near residual limbs, but early mistakes do not automatically propagate in time, so they can recover on frames where the stump is well visible. Consistent with this interpretation, intact joints such as knees and ankles remain in the 70+ AP range in our video benchmark, which is close to what comparable architectures report on PoseTrack under related settings (Liu et al., 2021b; He & Yang, 2024).

**Failure cases on InclusiveVidPose.** Figure 8 illustrates typical failure cases of DSTA on InclusiveVidPose. In these sequences the global pose is roughly plausible and the temporal aggregation produces visually coherent predictions across frames, yet the residual endpoints and nearby joints are often placed at anatomically implausible locations relative to the clearly shortened limbs and prostheses in the ground truth. These examples support our quantitative finding that existing multi frame architectures, which are tuned for intact body benchmarks and fixed skeletons, do not yet capture residual limb anatomy even when trained on the same annotations. The fact that intact joints reach normal AP while residual endpoints remain much harder to localize reinforces our main claim that InclusiveVidPose provides a demanding but realistic testbed for developing temporal pose models that explicitly represent limb deficiencies.

# F    DISCUSSION AND FUTURE WORK

**Limitation**    Even with a careful multi-annotator process and video, telling an occluded limb from an absent limb is sometimes unclear. This can add small noise to labels and make training a bit harder. We also see that current pose models on our data sometimes give unreliable confidence for missing or prosthetic limbs. These points suggest clear next steps for better guidelines and better models.

**Opportunities for prosthesis-aware modeling.**    Beyond benchmarking, our annotations are designed to support inclusive model design. First, while the current keypoint schema focuses on human anatomy and does not include prosthesis-tip joints, our pixel-level masks and per-limb prosthesis status naturally enable future work that introduces prosthesis end-effectors for applications requiring precise contact modeling with the environment (e.g., gait analysis or object manipulation). Second, the per-limb prosthesis status can be used by future pose estimators through conditional keypoint heads, adaptive skeleton graphs, or curriculum training schemes that explicitly condition on prosthesis presence.

**Privacy-preserving synthetic data augmentation.**    In future work, we will explore the use of synthetic data to protect participant privacy while enriching the InclusiveVidPose dataset. Real-world videos may raise privacy and ethical concerns and can be difficult or impractical to collect for rare limb-deficiency cases. Crucially, the success of any synthetic-data pipeline depends on a high-quality base of real examples. In our case, our InclusiveVidPose corpus provides the anatomical diversity and motion variability needed to guide realistic synthesis. To overcome the scarcity of certain amputation scenarios, we will generate motion sequences of limb-deficient subjects by combining advanced human-body simulators with generative adversarial networks. These synthetic clips will span uncommon amputation levels, diverse prosthesis designs, and challenging viewpoints or occlusions. We will then integrate them with our real footage. Finally, domain-adaptation and style-transfer techniques will harmonise appearance and temporal dynamics across synthetic and real data, ensuring that models trained on the hybrid dataset generalize robustly to in-the-wild videos.

---

**Algorithm 1** LiCC: Limb-specific Confidence Consistency

---

**Require:** Ground-truth annotations $G$, keypoint predictions $D$, number of keypoints $K$, rules $\mathcal{R}$
**Ensure:** LiCC score

 1: $c \leftarrow 0$                                              ▷ correct comparisons
 2: $N \leftarrow 0$                                                ▷ total comparisons
 3: **for all** $(g, d)$ in $\text{zip}(G, D)$ **do**
 4:      $\text{gt} \leftarrow \text{reshape}(g.keypoints, K, 3)$
 5:      $\text{pred} \leftarrow \text{reshape}(d.keypoints, K, 3)$
 6:      **for all** rule $\in \mathcal{R}$ **do**
 7:          $i \leftarrow \text{rule.main\_kp\_idx}$
 8:          $\mathcal{C} \leftarrow \text{rule.compare\_kp\_indices}$
 9:          $v \leftarrow \text{gt}[i, 2]$                                 ▷ ground-truth visibility
10:          **if** $v \in \{1, 2\}$ **then**
11:              $N \leftarrow N + 1$
12:              $m \leftarrow \text{pred}[i, 2]$
13:              $\text{mc} \leftarrow \{\, \text{pred}[j, 2] : j \in \mathcal{C}, 0 \leq j < K \,\}$
14:              $q \leftarrow \max(\text{mc})$
15:              **if** $m > q$ **then**
16:                  $c \leftarrow c + 1$
17:              **end if**
18:          **end if**
19:      **end for**
20: **end for**
21: **return** $\dfrac{c}{N}$ **if** $N > 0$ **else** 0

---

**Mutlimodal data for individuals with limb deficiencies.** While the current dataset focuses on RGB video frames with 2D keypoint and segmentation annotations, extending the data modalities could unlock richer biomechanical and temporal analyses. In particular, incorporating depth maps or multi-view camera streams would provide 3D pose ground truth and enable reconstruction-based methods to handle occlusions more robustly. Additionally, fusing inertial measurement unit (IMU) data from wearable sensors could offer complementary kinematic signals for smoother tracking of residual limbs. Finally, exploring audio-visual cues, for example, gait-associated sounds or speech commands, may facilitate multimodal models that better understand the context and intent of individuals with limb deficiencies during natural activities.

## G    IMPLEMENTATION DETAILS

**Experimental Setup** All experiments were run on a workstation equipped with two NVIDIA RTX 4090 GPUs and an AMD Ryzen Threadripper PRO 3955WX CPU. The system software environment is Ubuntu 24.04.2 LTS, Python 3.11.11, PyTorch 2.5.1 with CUDA 12.4. Our implementation is based on the MMPose library[1]). For data preprocessing, augmentation, and training hyperparameters, we used the default settings of each method as provided by MMPose. All training configurations, including exact config files and command-line instructions, have been released in the ⌂ InclusiveVidPose.

**Limb-specific Confidence Consistency(LiCC) Pseudocode Explanation** As shown in Algorithm 1, we denote $G$ as the set of all ground-truth annotations, $D$ as the set of corresponding pose estimation results, $K$ as the total number of keypoints, and $\mathcal{R}$ as the collection of mutual-exclusion

---

[1]https://github.com/open-mmlab/mmpose

rules. In our case,

$$\mathcal{R} = \Big\{ \underbrace{(7, \{17\}), \ (9, \{17, 19\}), \ (17, \{7, 9\}), \ (19, \{9\}),}_{\text{Left Upper Extremity}}$$

$$\underbrace{(8, \{18\}), \ (10, \{18, 20\}), \ (18, \{8, 10\}), \ (20, \{10\}),}_{\text{Right Upper Extremity}}$$

$$\underbrace{(13, \{21\}), \ (15, \{21, 23\}), \ (21, \{13, 15\}), \ (23, \{15\}),}_{\text{Left Lower Extremity}}$$

$$\underbrace{(14, \{22\}), \ (16, \{22, 24\}), \ (22, \{14, 16\}), \ (24, \{16\})}_{\text{Right Lower Extremity}} \Big\},$$

where each tuple $(i, \{j, \dots\})$ enforces that, whenever keypoint $i$ is visible in the ground truth, its predicted confidence must strictly exceed the maximum confidence among the mutually-exclusive indices $\{j, \dots\}$. The algorithm maintains two counters, $c$ for correct comparisons and $N$ for total comparisons. For each pair $(g, d) \in \texttt{zip}(G, D)$, we skip if $g$ is empty; otherwise we reshape $g.keypoints$ into $\text{gt} \in \mathbb{R}^{K \times 3}$ (visibility in column 3) and $d.keypoints$ into $\text{pred} \in \mathbb{R}^{K \times 3}$, extracting its third column as the predicted confidence vector. We then iterate over every $(i, \mathcal{C}) \in \mathcal{R}$: ensure $i \in [0, K)$, read visibility $v = \text{gt}[i, 2]$, and if $v \in \{1, 2\}$ increment $N$, set $m = \text{pred}[i]$, gather $\{\text{pred}[j] \mid j \in \mathcal{C}\}$ into mc, skip if empty, else let $q = \max(\text{mc})$ and if $m > q$ increment $c$. Finally, we return $c/N$ when $N > 0$, or 0 otherwise. This LiCC metric thus measures how consistently a visible keypoint's confidence surpasses that of all its mutually exclusive counterparts.

## H    TERMS OF USE

### H.1    ANNOTATIONS AND WEBSITE

The annotations and all website content for the InclusiveVidPose dataset are (c) 2025 InclusiveVidPose Consortium, and are released under the Creative Commons BY-NC-SA 4.0 license ©.

### H.2    INFRINGEMENT

If you believe any material in this dataset infringes your rights, please contact the authors. We will promptly review and remove the infringing content.

### H.3    VIDEOS AND FRAMES

All videos referenced in this dataset are publicly available YouTube videos or YouTube Shorts. We do not host or distribute the video files or extracted frames, instead, each annotation record includes the corresponding publicly accessible YouTube URL. Any use of these videos or frames (e.g., viewing, downloading, processing, or redistributing) must fully comply with YouTube's Terms of Service (https://www.youtube.com/t/terms) as well as any additional restrictions imposed by the original content creators or channels. Downstream users assume sole responsibility for ensuring that their use of this material conforms to all applicable copyright, privacy, and publicity laws.

Under both Australian and U.S. copyright law, academic researchers may, under certain conditions, use publicly available media, including YouTube videos, for research and study. In Australia, the Copyright Act 1968 provides fair dealing exceptions that permit the use of copyright material for research or study where the use is fair. In the United States, Section 107 of the Copyright Act codifies fair use and explicitly identifies scholarship and research as purposes that can qualify under this doctrine. Our own collection and internal use of source videos falls within these research-focused exceptions; however, this dataset only distributes derived annotations (e.g., keypoints and segmentation masks), not the underlying video content. Users remain responsible for ensuring that any use of the source videos complies with applicable copyright law and the terms of service of the hosting platforms.

## H.4    ETHICS ADDENDUM

By accessing, downloading, or using the InclusiveVidPose dataset (the "Dataset"), any downstream user ("You") must agree to and abide by the following additional ethical obligations, which supplement the Dataset's base license terms:

- **Anonymization Standards**

  You may not distribute, publish, or otherwise reveal any video frames or metadata in a form that could identify any individual, minor or adult, either directly or by inference.

- **Prohibition on Re-Identification**

  You shall not attempt, by any means, to re-identify any person depicted in the Dataset.

  You shall not augment the Dataset with external data or use deanonymization algorithms to recover or expose personal identities.

- **Non-Commercial, Ethical Research Use Only**

  You shall use the Dataset solely for non-commercial, research-oriented activities that comply with all applicable laws, institutional review board (IRB) approvals or exemptions, and recognized ethical norms for human-subjects research.

  Any form of commercial exploitation, product development, or profit-driven application of the Dataset is expressly prohibited without the prior written consent of the Dataset's custodians.

- **Breach and Remedies**

  Violation of any provision in this Addendum constitutes a material breach of your rights to use the Dataset, and may result in immediate termination of your license and access privileges. The Dataset custodians reserve the right to pursue any and all available legal and equitable remedies in the event of non-compliance.

## H.5    DATA USE AGREEMENT (DUA)

As a prerequisite for accessing the InclusiveVidPose dataset, all prospective users must complete a Data Use Agreement and submit it for review. Each agreement will be examined by the dataset administrators to ensure that proposed uses comply with our ethical and legal guidelines. Only after the DUA has been approved will access credentials be issued. A scanned copy of the signed agreement is provided in the supplementary materials.

