# OpenReview forum: "InclusiveVidPose: Bridging the Pose Estimation Gap for Individuals with Limb Deficiencies in Video-Based Motion"
_ICLR.cc/2026/Conference — ICLR 2026 Poster_

### Official Review · Reviewer_22nG · 2025-10-30

**Soundness:** 4
**Presentation:** 4
**Contribution:** 4
**Rating:** 8
**Confidence:** 5

**Summary:**

InclusiveVidPose introduces the first large scale video based human pose estimation dataset focused on individuals with limb deficiencies, addressing the lack of representation in existing benchmarks which assume anatomically intact bodies . The dataset contains 313 videos totaling 327k frames featuring nearly 400 participants with amputations, congenital limb differences, and prostheses, and it adds eight residual limb end keypoints to accurately model altered anatomy . Each frame includes pose keypoints, segmentation masks, bounding boxes, subject tracking, and per limb prosthesis status collected with expert guidance. Experiments show that state of the art human pose estimation models frequently predict anatomically impossible joints in the presence of missing or prosthetic limbs, motivating a new evaluation metric called Limb specific Confidence Consistency (LiCC) to measure confidence calibration under anatomical absence. InclusiveVidPose provides a challenging benchmark intended to drive research toward inclusive and robust pose estimation for rehabilitation, health assessment, and assistive technologies.

**Strengths:**

- The paper presents a dataset of approximately 327,000 video frames featuring various types of limb deficiencies. It also introduces a new metric, called Limb-specific Confidence Consistency (LiCC), designed to help evaluate whether pose-estimation models can tell the difference between intact limbs and those that are residual or absent.
- As shown in Appendix D, the AP scores on residual endpoints are lower than on standard joints. Moreover, while adding COCO during training enhances performance on standard joints, it deteriorates performance on residual limbs. This highlights a research gap in how to effectively combine the two without harming performance on either group, suggesting a valuable direction for future work.
- The proposed dataset is in a video modality, which is crucial for individuals with amputations because per-frame information can help disambiguate occluded limbs from truly absent ones. Current 2D pose baselines rely on image-based datasets and often perform poorly on metrics such as LiCC. This suggests that models incorporating video modality are needed to reduce such ambiguity.
- A comprehensive evaluation of the dataset is conducted using recent 2D pose estimators, including YOLOX-Pose, DEKR, ViPNAS, Swin Transformer, RTMPose, and ViTPose, to analyze their behavior on the proposed dataset using COCO metrics in addition to the newly introduced LiCC metric. Furthermore, parallel experiments are performed to examine how incorporating COCO into training affects performance on both the COCO and InclusiveVidPose datasets.

**Weaknesses:**

- Although the paper highlights the ambiguity between occluded and absent limbs when annotating data, where annotators use temporal cues to disambiguate, the baseline comparisons are all image based. It would have been beneficial if a simple video baseline had been introduced to demonstrate how temporal cues can help the model resolve this ambiguity and how future work could further improve video based methods.

**Questions:**

- At line 309, the “annotation team training” field is empty. Is there any missing information?

**Details Of Ethics Concerns:**

As stated at 742, there is a risk of re-identification that can compromise personal privacy. The work uses data use agreement to forbid such use cases but still there is a risk of doing it.

---

> ### Author Response · Authors · 2025-11-24
> **Response to Reviewer 22nG**
>
> We thank the reviewer for the very positive and detailed review and for the strong support for acceptance. We address the remaining concerns below.
>
> ---
>
> ### W1. No video baseline
>
> > The dataset is video-based, so it would be helpful to see at least one simple video (temporal) baseline to understand how much temporal information helps, especially for disambiguating occluded vs missing limbs.
> >
>
> **Table R3: PoseTrack-style keypoint AP on InclusiveVidPose.**
>
> |  | Head | Shoulder | Elbow | Wrist | Hip | Knee | Ankle | ArmUp | ArmLow | LegUp | LegLow | Mean |
> | --- | --- | --- | --- | --- | --- | --- | --- | --- | --- | --- | --- | --- |
> | DCPose | 28.13 | 72.02 | 69.68 | 79.34 | 73.24 | 72.44 | 72.90 | 1.63 | 0.17 | 12.21 | 16.03 | 43.20 |
> | DSTA | 28.92 | 72.17 | 70.44 | 81.93 | 72.70 | 71.93 | 71.99 | 0.64 | 0.01 | 14.28 | 17.27 | 43.65 |
>
> **Response.** Thank you for this suggestion. Following your comment, we have added a multi-frame pose estimation benchmark. Table R3 now reports PoseTrack-style keypoint AP on InclusiveVidPose for two representative video pose estimators, DCPose [1] and DSTA [2]. Both models keep relatively high AP on standard joints (shoulders, elbows, wrists, hips, knees, and ankles are all in the low–mid 70s, with DSTA slightly stronger on several joints), while head AP is much lower (around 28 AP) due to occlusion, cropping, and motion blur. In contrast, AP on the four residual-limb groups is low: ArmUp and ArmLow are close to zero for both methods, and LegUp/LegLow remain in the low–mid teens. The mean AP over all keypoints only increases from 43.2 (DCPose) to 43.7 (DSTA), showing that modest gains on intact joints do not close the large gap on residual-limb regions. These results indicate that current video pose estimators, which are trained and tuned on intact-body datasets, still do not transfer to our residual-limb keypoints. We have added this multi-frame experiment and its analysis to the main paper in the revised submission.
>
> [1] Liu, Z., Chen, H., Feng, R., Wu, S., Ji, S., Yang, B., & Wang, X. (2021). Deep dual consecutive network for human pose estimation. In *Proceedings of the IEEE/CVF conference on computer vision and pattern recognition* (pp. 525-534).
> [2] He, J., & Yang, W. (2024). Video-based human pose regression via decoupled space-time aggregation. In *Proceedings of the IEEE/CVF conference on computer vision and pattern recognition* (pp. 1022-1031).
>
> ---
>
> ### Q1. Annotation team training field
>
> > At line 309, the “annotation team training” field is empty. Is there any missing information?
> >
>
> **Response.** Thank you for pointing this out. We have corrected this oversight in the revised submission and filled in the annotation team training field.
>
> ---
>
> ### E1. Ethics and privacy risks
>
> > Concern about re-identification and privacy risks when working with web videos featuring people with disabilities.
> >
>
> **Response.** We appreciate this concern and we take it seriously. The dataset is collected and used under approval from the institutional Human Research Ethics Committee (HREC). The public release only contains anonymized annotations (keypoints, segmentation masks, bounding boxes, tracking IDs, and prosthesis status) together with video URLs, so the original content remains under the control and terms of the hosting platforms and rights holders. Access to the annotations is gated by a Data Use Agreement that restricts use to non-commercial research and prohibits re-identification, face recognition, or other non-research uses.
>
> We also acknowledge that, as you note, some residual risk of re-identification remains whenever people appear in online videos. Because our task involves facial keypoints, we do not currently blur faces in the underlying source videos, but we plan to explore releasing a future version of the dataset with face blurring and/or reduced facial detail to further lower this risk while preserving pose-estimation utility.

---

> > ### Comment · Reviewer_22nG · 2025-11-26
> >
> > Thank you for inclusion of DCPose and DSTA.
> > 1. Is there any reason why AP for these models are significantly less than image models?
> > 2. Are these models trained on InclusiveVidPose and also tested on InclusiveVidPose?

---

> > > ### Author Response · Authors · 2025-11-27
> > > **Response to Reviewer 22nG**
> > >
> > > Thank you for this follow up and for asking about the DCPose and DSTA results.
> > >
> > > > Are these models trained on InclusiveVidPose and also tested on InclusiveVidPose?
> > > >
> > >
> > > **Response.** Yes, both DCPose and DSTA are trained on the InclusiveVidPose training split and evaluated on the InclusiveVidPose validation and test splits. We adapt their heads to the 25 keypoints and use the PoseTrack style evaluation protocol on multi-frame pose estimation.
> > >
> > > ---
> > >
> > > > Is there any reason why AP for these models are significantly less than image models?
> > > >
> > >
> > > **Response.** For the head joints, there are two dataset related factors that make the task harder than in PoseTrack style benchmarks. First, many InclusiveVidPose sequences center on the torso and residual limbs, for example rehabilitation training or prosthesis demonstrations, so the head is often close to the crop boundary or partially out of frame. This framing differs from typical PoseTrack footage, where heads are usually clear. Second, facial keypoints are annotated in fewer instances than most torso and leg joints. Each head joint appears only on the order of one hundred thousand labeled instances in our video annotations, which is roughly about half of the supervision frequency that standard lower body joints receive. This means that all models, including DCPose and DSTA, see noticeably fewer positive examples for the head than for knees, hips, and ankles during training.
> > >
> > > The larger gap on residual endpoints comes mainly from how current video pose models are designed. DCPose and DSTA are built to exploit temporal continuity in videos where every subject has a complete skeleton. They always predict a fixed set of joints for each person in every frame, and they aggregate neighboring frames to smooth short term occlusions, blur, or defocus. On PoseTrack this assumption holds, because there are no limb deficiencies and occlusions are usually temporary.  In InclusiveVidPose, residual limbs are structural rather than temporary. Many participants have only one or two residual endpoints and the rest of the limb is truly absent. When a model that assumes a full limb sees such a sequence, it tends to hallucinate a complete arm or leg and gradually pull the residual endpoint toward a typical wrist or ankle location. Once this happens near the beginning of a clip, the temporal module propagates and stabilizes the wrong configuration across many frames, so residual endpoints follow a smooth but systematically shifted trajectory, and they receive low AP under per frame evaluation.
> > >
> > > The image baselines instead make independent predictions at each frame. They still struggle near residual limbs, but early mistakes do not automatically propagate in time, so they can recover on frames where the stump is well visible. Consistent with this interpretation, intact joints such as knees and ankles remain in the 70+ AP range in our video benchmark, which is close to what comparable architectures report on PoseTrack under related settings.  To make these behaviors more concrete, we add a figure in the supplementary material that illustrates typical failure cases of DSTA on InclusiveVidPose. This figure shows that even when the global pose is roughly plausible, the predicted residual endpoints and nearby joints are often placed at anatomically implausible locations relative to the clearly shortened limbs and prostheses in the ground truth.
> > >
> > > Overall, we view the lower AP of DCPose and DSTA on residual limb endpoints as evidence that InclusiveVidPose exposes a new and important challenge. Existing multi-frame pose architectures, which are tuned for intact body benchmarks and fixed skeletons, do not yet capture residual limb anatomy, even when they are trained on the same annotations. The fact that intact joints reach normal AP while residual endpoints remain much harder to localize supports our main claim that InclusiveVidPose provides a demanding but realistic testbed for developing pose models that explicitly represent limb deficiencies.

---

### Official Review · Reviewer_uQTG · 2025-10-31

**Soundness:** 3
**Presentation:** 3
**Contribution:** 3
**Rating:** 6
**Confidence:** 2

**Summary:**

This paper presents InclusiveVidPose, the first large-scale video-based human pose estimation (HPE) dataset designed specifically for individuals with limb deficiencies. The dataset contains 313 videos (≈327k frames) featuring 398 participants with various amputation and congenital limb-difference types. It introduces eight new “residual-limb end” keypoints extending the COCO 17-keypoint schema to better model missing or prosthetic limbs. Each frame is annotated with segmentation masks, bounding boxes, tracking IDs, and per-limb prosthesis status, verified under expert para-athletic supervision. The paper also proposes a new evaluation metric, Limb-specific Confidence Consistency (LiCC), which measures whether a model maintains coherent confidence calibration between existing and absent limbs. Extensive benchmarks with 12 existing pose models (e.g., ViTPose, Swin, HRNet, YOLOPose) reveal that current systems generalize poorly to limb-difference anatomies, often predicting anatomically implausible joints or over-confident scores on missing limbs.

**Strengths:**

1) The dataset is well-curated and technically rigorous: (i) Video-based rather than static image design allows temporal disambiguation between occluded and absent limbs. (ii) Multi-modal annotation (pose, mask, prosthesis status, tracking) ensures reusability across research domains (pose, segmentation, tracking, rehabilitation). It will be a great contribution to the community.

2) LiCC is a conceptually sound and practically useful metric to quantify anatomical plausibility and confidence consistency — an important step toward calibrated pose estimation models. It provides a tangible way to evaluate inclusiveness beyond accuracy.

3) In general, the paper addresses a genuine fairness gap in pose estimation research, highlighting the underrepresentation of hundreds of millions of people with limb differences.

**Weaknesses:**

1) While the dataset and metric are valuable and they are good contributions, the methodological side (pose estimation models) remains entirely benchmark-based. The paper would benefit from proposing a baseline model explicitly optimized for inclusive anatomy (e.g., deformable skeleton graph, conditional keypoint masking).

2) Despite the dataset being video-based, all experiments are single-frame. The paper defers temporal modeling to “future work,” missing an opportunity to demonstrate how temporal priors improve residual-limb estimation.

3) Given the high variability in anatomy and camera conditions, annotation uncertainty modeling (e.g., probabilistic keypoints or confidence intervals) could be beneficial. The paper treats annotations as deterministic. That is potentially a good future work worth exploring further.

**Questions:**

1) Since the dataset is video-based, have the authors tested temporal models like PoseTrack transformers or 3D CNNs? How does motion continuity affect missing-limb recognition?

2) What happens if models are pre-trained on COCO and fine-tuned on InclusiveVidPose? Is catastrophic forgetting an issue for intact-body keypoints? Just curious about the outcome.

3) Just wondering- could segmentation masks be used to guide pose estimation, e.g., via conditional keypoint visibility modeling?

---

> ### Author Response · Authors · 2025-11-24
> **Response to Reviewer uQTG (1/2)**
>
> We thank the reviewer for the careful review and for recognizing the value of the dataset and the LiCC metric. We address the main points below.
>
> ---
>
> ### W1. No inclusive-specific baseline model
>
> > While the dataset and metric are valuable, the modeling side remains purely benchmark-based; a baseline that explicitly targets inclusive pose estimation would strengthen the paper.
> >
>
> **Response.** We agree that inclusive, prosthesis-aware architectures are important, and we see this as exactly the gap our work aims to surface. Individuals with limb deficiencies are almost absent from existing pose benchmarks, so current pose estimators are rarely designed or evaluated with them as a primary target. In this submission we therefore focus on providing a curated dataset, the LiCC metric, and a standardized benchmark using unmodified off-the-shelf models, so that the current performance gap on disability-related regions becomes clear and reproducible. In the revised paper we will explicitly state this scope and emphasize that InclusiveVidPose is intended as a foundation for future inclusive models, encouraging the community to develop prosthesis-aware pose architectures that directly address the challenges we expose.
>
> ---
>
> ### W2/Q1. No video baseline on a video dataset
>
> > The dataset and annotations are video-based, yet all baselines are single-frame models; including a simple temporal baseline would show how much temporal information helps.
> >
>
> **Table R3: PoseTrack-style keypoint AP on InclusiveVidPose.**
>
> |  | Head | Shoulder | Elbow | Wrist | Hip | Knee | Ankle | ArmUp | ArmLow | LegUp | LegLow | Mean |
> | --- | --- | --- | --- | --- | --- | --- | --- | --- | --- | --- | --- | --- |
> | DCPose | 28.13 | 72.02 | 69.68 | 79.34 | 73.24 | 72.44 | 72.90 | 1.63 | 0.17 | 12.21 | 16.03 | 43.20 |
> | DSTA | 28.92 | 72.17 | 70.44 | 81.93 | 72.70 | 71.93 | 71.99 | 0.64 | 0.01 | 14.28 | 17.27 | 43.65 |
>
> **Response.** Thank you for this suggestion. Following your comment, we have added a multi-frame pose estimation benchmark. Table R3 now reports PoseTrack-style keypoint AP on InclusiveVidPose for two representative video pose estimators, DCPose [1] and DSTA [2]. Both models keep relatively high AP on standard joints (shoulders, elbows, wrists, hips, knees, and ankles are all in the low–mid 70s, with DSTA slightly stronger on several joints), while head AP is much lower (around 28 AP) due to occlusion, cropping, and motion blur. In contrast, AP on the four residual-limb groups is low: ArmUp and ArmLow are close to zero for both methods, and LegUp/LegLow remain in the low–mid teens. The mean AP over all keypoints only increases from 43.2 (DCPose) to 43.7 (DSTA), showing that modest gains on intact joints do not close the large gap on residual-limb regions. These results indicate that current video pose estimators, which are trained and tuned on intact-body datasets, still do not transfer to our residual-limb keypoints. We have added this multi-frame experiment and its analysis to the main paper in the revised submission.
>
> [1] Liu, Z., Chen, H., Feng, R., Wu, S., Ji, S., Yang, B., & Wang, X. (2021). Deep dual consecutive network for human pose estimation. In *Proceedings of the IEEE/CVF conference on computer vision and pattern recognition* (pp. 525-534).
> [2] He, J., & Yang, W. (2024). Video-based human pose regression via decoupled space-time aggregation. In *Proceedings of the IEEE/CVF conference on computer vision and pattern recognition* (pp. 1022-1031).
>
> ---
>
> ### W3. Annotation uncertainty modeling
>
> > Given the high variability in anatomy and camera conditions, annotation uncertainty modeling (e.g., probabilistic keypoints or confidence intervals) could be beneficial. The paper treats annotations as deterministic.
> >
>
> **Response.** We agree that uncertainty-aware annotations and models would be valuable, especially under occlusion and for people who consistently wear prostheses. To reduce ambiguity, annotators always inspect short video clips (rather than single frames) when placing residual endpoints. They follow written guidelines that are calibrated together with an IPC para-athletics classifier. Frames that remain undecidable after review are treated rather than forced into arbitrary labels. This aligns with our view that probabilistic keypoints and uncertainty-aware heads are a natural next step on top of InclusiveVidPose, especially for cases with persistent prosthesis use or heavy occlusion.

---

> > ### Author Response · Authors · 2025-11-24
> > **Response to Reviewer uQTG (2/2)**
> >
> > ### Q2. COCO pretraining and catastrophic forgetting
> >
> > > What happens if models are pre-trained on COCO and then trained on InclusiveVidPose? Is catastrophic forgetting an issue for intact-body keypoints?
> > >
> >
> > **Response.** We first clarify that all pose estimation models in Table 2 are initialized from their official COCO-pretrained weights before training on InclusiveVidPose and/or COCO, and we explicitly state this in the revised paper.
> >
> > **Table R4: ViTPose-H on InclusiveVidPose with and without COCO pretraining.** “InclusiveVidPose* → InclusiveVidPose” trains ViTPose-H **from scratch** on InclusiveVidPose. “InclusiveVidPose → InclusiveVidPose” starts from **COCO-pretrained weights** and then trains on InclusiveVidPose.
> >
> > |  | InclusiveVidPose* -> InclsuiveVidPose |  |  |  |  |  | InclusiveVidPose -> InclsuiveVidPose |  |  |  |  |  |
> > | --- | --- | --- | --- | --- | --- | --- | --- | --- | --- | --- | --- | --- |
> > |  | AP | AP50 | AP75 | AR | AR50 | AR75 | AP | AP50 | AP75 | AR | AR50 | AR75 |
> > | 8 | 82.4 | 90.1 | 85.4 | 85.2 | 91.6 | 87.2 | 84.2 | 92.2 | 86.3 | 86.7 | 93.1 | 88.0 |
> > | 17 | 85.2 | 89.6 | 86.3 | 86.6 | 90.9 | 87.4 | 86.7 | 90.8 | 87.5 | 87.9 | 91.8 | 88.6 |
> > | 25 | 84.8 | 90.6 | 86.3 | 86.3 | 91.1 | 89.4 | 86.3 | 90.8 | 87.5 | 87.6 | 91.8 | 88.7 |
> >
> > As shown Table R4, we add a controlled comparison for ViTPose-H. In the “InclusiveVidPose* → InclusiveVidPose” setting, ViTPose-H trains from scratch on InclusiveVidPose only. In “InclusiveVidPose → InclusiveVidPose”, the same architecture starts from COCO-pretrained weights and then trains on InclusiveVidPose. COCO pretraining brings consistent gains across all keypoint sets: AP/AR improve from 82.4/85.2 to 84.2/86.7 on the 8 residual endpoints, from 85.2/86.6 to 86.7/87.9 on the 17 standard joints, and from 84.8/86.3 to 86.3/87.6 on the full 25-keypoint schema. The improvements are similar in magnitude for residual endpoints and standard joints, which suggests that pretraining on a large intact-body dataset provides a useful generic pose prior without harming residual-endpoint localization.
> >
> > Together with our joint-training results, these trends indicate that COCO mainly acts as a strong initialization that stabilizes common body structure, and we do not observe catastrophic forgetting of intact-body keypoints when training on InclusiveVidPose. The numerical results and analysis are included in Supplementary Section D as Table 4.
> >
> > ---
> >
> > ### Q3. Using segmentation masks to guide pose
> >
> > > Using segmentation masks to guide pose estimation, especially for disability-related cases, could be beneficial.
> > >
> >
> > **Response.** We agree that segmentation could be a useful cue for inclusive pose estimation. However, this would require **body-part–level parsing masks** rather than the coarse person-level masks that are common in existing datasets. Person-level silhouettes only indicate where a human is, but they do not distinguish between intact limbs, residual limbs, and prosthetic components, so they cannot reliably resolve the limb-level ambiguities we study. In contrast, our InclusiveVidPose annotations focus on keypoints; combining them with body parsing masks is an attractive direction for future work.

---

### Official Review · Reviewer_yopn · 2025-11-01

**Soundness:** 4
**Presentation:** 4
**Contribution:** 4
**Rating:** 8
**Confidence:** 4

**Summary:**

The paper introduces InclusiveVidPose, a large-scale video-based human pose estimation dataset focused on individuals with limb deficiencies. The paper argues that existing models are trained only on able-bodied individuals with full keypoints, which results in failing to generalize to this population. To address this, the authors have annotated a large amount of data with bounding boxes, segmentation masks, and keypoints. Additionally, the paper introduces an extended 25-keypoint schema for people with limb deficiencies, building on the 17 COCO standard. Lastly, the paper proposes LiCC, a percentage-based metric to evaluate how confident a model predicts limb deficiencies between mutually exclusive joints'. The paper presents a comprehensive set of experiments on recent HPE models and highlights the research gap this dataset addresses.

**Strengths:**

1. The problem is very well-motivated, and the paper addresses a clear gap in research. The paper presents all the tools necessary for future research in this domain, including a high-quality dataset, annotations, evaluation tools, and baselines.
2. The dataset statistics show a well-balanced distribution of the population.
3. The experimental evaluations are very comprehensive. The low LiCC score also supports the paper's claim about the necessity of this research.
4. The paper is very well-written and uses appropriate words for its arguments and to report its findings.

**Weaknesses:**

1. The results show that by training on the introduced dataset, the performance of some of the models drops. The paper notes that this is due to the size of the models, larger output space, and their architectural optimization for the COCO dataset. However, this can also occur due to annotation inconsistencies across datasets. However, the paper lacks any quantitative evaluation of the annotation consistency. While the qualitative reports and supplementary materials include high-quality labels, additional consistency analysis would benefit the paper.
2. There's a typo on line 515, where "sp lits" should be "splits"
3. The lack of annotation for limb prosthesis end-joint coordinates is a minor weakness in this paper, as many HPE applications rely on how humans interact with their surroundings (e.g., walking, grabbing objects). Marking these end nodes as invisible in the annotations could hinder their performance.
4. While the dataset labels the per-limb prosthesis status, it does not use it in the models. Still, this is not a significant weakness, and future work can build on it.

**Questions:**

1. As I mentioned above, some analysis of consistency/annotation quality seems to be necessary. Table 2 already provides an exhaustive overview of the results. However, adding a "COCO -> InclusiveVidPose" column and comparing it with "COCO -> COCO" can reveal the distribution shift between the two datasets. This analysis can also be broken down to limb deficiency types for more insight.

**Details Of Ethics Concerns:**

I am not an expert, but relying on YouTube videos can cause legal issues in some countries that should be investigated if this paper meets those standards.

---

> ### Author Response · Authors · 2025-11-24
> **Response to Reviewer yopn (1/2)**
>
> We thank the reviewer for the very positive and constructive review and for rating soundness, contribution, and presentation as excellent. Below we respond to each point in turn.
>
> ---
>
> ### W1/Q1: Annotation consistency and COCO → InclusiveVidPose
>
> > The performance differences between COCO and InclusiveVidPose may partly come from annotation inconsistencies. It would be helpful to add a COCO → InclusiveVidPose evaluation column.
> >
>
> **Response.** Thank you for suggesting to add a COCO → InclusiveVidPose column and to compare it with *COCO → COCO*. In the revised Table 2, we now include this setting. All models are trained on COCO, and we evaluate them on InclusiveVidPose using a 17-keypoint version of InclusiveVidPose that keeps only the COCO joints and ignores the 8 residual endpoints. We also report *COCO → COCO* under the same 17-keypoint label space.
>
> **Table R1: COCO-pretrained models on COCO and InclusiveVidPose with a 17-keypoint protocol. We report AP and AR for COCO → InclusiveVidPose and COCO → COCO using the shared COCO 17-keypoint label space.**
>
> | Method | Backbone | Input Size | COCO |  | COCO |  |
> | --- | --- | --- | --- | --- | --- | --- |
> |  |  |  | InclusiveVidPose |  | COCO |  |
> |  |  |  | AP | AR | AP | AR |
> | Yoloxpose | YoloxPose-T | 416x416 | 74.0 | 91.1 | 52.6 | 57.1 |
> |  | YoloxPose-S | 640x640 | 78.4 | 93.3 | 64.1 | 68.2 |
> |  | YoloxPose-M | 640x640 | 79.5 | 94.4 | 69.5 | 89.9 |
> |  | YoloxPose-L | 640x640 | 80.0 | 94.9 | 71.2 | 90.1 |
> | DEKR | HRNet-w32 | 512x512 | 55.8 | 93.2 | 68.6 | 79.5 |
> |  | HRNet-w48 | 640x640 | 53.4 | 93.4 | 71.4 | 76.2 |
> | ViPNAS | MobileNetV3 | 256x192 | 70.2 | 72.9 | 69.5 | 75.5 |
> | Swin | Swin-T | 256x192 | 72.6 | 75.8 | 72.4 | 78.2 |
> |  | Swin-B | 256x192 | 77.3 | 80.2 | 73.7 | 79.4 |
> |  | Swin-B | 384x288 | 78.8 | 81.6 | 75.9 | 81.1 |
> |  | Swin-L | 256x192 | 77.1 | 79.9 | 74.3 | 79.8 |
> |  | Swin-L | 384x288 | 78.4 | 81.4 | 76.3 | 81.4 |
> | RTMPose | RTMPose-T | 256x192 | 66.5 | 68.8 | 68.2 | 73.6 |
> |  | RTMPose-S | 256x192 | 71.8 | 74.0 | 71.6 | 76.8 |
> |  | RTMPose-M | 256x192 | 75.2 | 77.7 | 74.6 | 79.5 |
> |  | RTMPose-L | 256x192 | 76.2 | 78.7 | 75.8 | 80.6 |
> | ViTPose | ViT-S | 256x192 | 72.5 | 75.2 | 73.9 | 79.2 |
> |  | ViT-B | 256x192 | 77.4 | 80.0 | 75.7 | 81.0 |
> |  | ViT-L | 256x192 | 79.9 | 82.5 | 78.2 | 83.4 |
> |  | ViT-H | 256x192 | 81.5 | 83.8 | 78.8 | 83.9 |
>
> As shown in Table R1, many methods show lower AP or AR and their ranking changes, which indicates that InclusiveVidPose constitutes a different and more challenging test distribution. At the same time, strong models such as ViTPose-H and YOLOX-Pose-L still achieve high scores in the *COCO →  InclusiveVidPose* setting (e.g., 81.5 AP / 83.8 AR and 80.0 AP / 94.9 AR), and they remain among the top-performing architectures in both columns. Since the training data and the evaluated 17 joints are identical, this pattern is better explained by a real distribution shift between COCO and InclusiveVidPose combined with different cross-dataset robustness across architectures. It is less likely to be caused by inconsistent keypoint annotations in InclusiveVidPose.
>
> **Table R2: ViTPose-H performance on 17 COCO keypoints by limb-deficiency group.** COCO-style AP and AR on InclusiveVidPose using the 17 shared keypoints, grouped into Arm Left, Arm Right, Leg Left, Leg Right residual cases and Intact.
>
> |  | AP | AP50 | AP75 | AR | AR50 | AR75 |
> | --- | --- | --- | --- | --- | --- | --- |
> | Intact | 75.7 | 84.1 | 76.4 | 81.5 | 88.1 | 83.0 |
> | Arm Left | 84.7 | 88.4 | 85.9 | 86.6 | 89.9 | 87.4 |
> | Arm Right | 82.3 | 86.1 | 82.9 | 86.2 | 89.7 | 86.8 |
> | Leg Left | 92.3 | 95.9 | 92.9 | 93.0 | 96.1 | 93.7 |
> | Leg Right | 87.5 | 91.9 | 88.6 | 88.7 | 93.0 | 89.1 |
>
> Furthermore, we analyze ViTPose-H on a 17-keypoint version of InclusiveVidPose and group clips into Arm Left, Arm Right, Leg Left, Leg Right, and Intact. As reported in Table R2, ViTPose-H achieves high AP and AR across all four limb-deficiency groups (about 82–85 AP for residual-arm clips and 87–92 AP for residual-leg clips), while the Intact group is lower (75.7 AP and 81.5 AR). For limb-deficiency clips, keypoints on the missing limb are annotated with visibility=0 and are ignored when COCO metrics are computed, so the evaluation focuses on the remaining visible standard joints; intact clips, in contrast, include more visible joints, including occluded and fast-moving points that are harder to localize. This pattern indicates that the shared 17 COCO joints are annotated consistently across limb-deficiency configurations, and that residual arms remain harder to localize than residual legs, likely due to stronger self-occlusion and articulation near the torso. Overall, these results support our claim that the main challenge lies in modeling residual-limb anatomy rather than in inconsistent placement of standard keypoints. This analysis and the corresponding results are included in Section D, Table 5 of the supplementary material.

---

> > ### Author Response · Authors · 2025-11-24
> > **Response to Reviewer yopn (2/2)**
> >
> > ### W2: Typo
> >
> > > “There is a ‘sp lits’ typo.”
> > >
> >
> > **Response.** Thank you for noticing this. We correct this typo and we carefully re-read the manuscript to remove similar issues in the revision.
> >
> > ---
> >
> > ### W3. Missing prosthesis end-joint keypoints
> >
> > > The lack of annotation for limb prosthesis end-joint coordinates is a minor weakness in this paper, as many HPE applications rely on how humans interact with their surroundings (e.g., walking, grabbing objects). Marking these end nodes as invisible in the annotations could hinder their performance.
> > >
> >
> > **Response.** We agree that end-effectors are important for many downstream tasks that reason about human–environment interaction. In this version of InclusiveVidPose, the keypoint schema focuses on **human anatomy** rather than device geometry.
> >
> > - Residual-limb endpoints are anatomical and **invariant to the specific prosthesis**, so they are stable landmarks across different devices and fittings.
> > - The distal tip of a prosthesis can vary across devices, sports attachments, and configurations. We therefore represent prosthesis geometry and contact using **segmentation masks** and per-limb prosthesis status instead of introducing new fixed keypoints at the prosthesis tip.
> >
> > We will clarify this design choice in the main text and state in the Limitations / Broader Impact sections that prosthesis-tip keypoints are a natural extension for applications that require precise contact modeling with the environment (for example, gait analysis or prosthetic hand–object interaction). More broadly, by explicitly representing residual limbs and prostheses in a large-scale video benchmark, we hope to draw attention to the systematic gap between people with and without limb differences in human pose and motion research, and to encourage future work that includes prosthesis end-joint keypoints and related signals so that models and datasets evolve toward a more inclusive community.
> >
> > ---
> >
> > ### W4. Per-limb prosthesis status not used in baselines
> >
> > > While the dataset labels the per-limb prosthesis status, it does not use it in the models. Still, this is not a significant weakness, and future work can build on it.
> > >
> >
> > **Response.** We agree that per-limb prosthesis status is a powerful signal, and we intentionally design the annotation so that future methods can exploit it.
> >
> > For this **Datasets and Benchmarks** submission we keep all baselines as **unmodified off-the-shelf pose models**. Our primary goal is to expose how current, widely used pose estimators behave on limb-difference anatomies under standard training, without introducing our own architecture that could shift the focus away from the dataset and the LiCC metric.
> >
> > We will make this scope choice explicit in the revision and add a short discussion of modeling directions that build on the per-limb prosthesis status, such as:
> >
> > - conditional keypoint heads that adapt predictions when a prosthesis is present,
> > - adaptive skeleton graphs that change limb connectivity based on prosthesis status,
> > - curriculum or staged training strategies that condition on prosthesis presence to gradually introduce limb-difference cases.
> >
> > We appreciate this suggestion because it aligns with our broader goal: we hope that by releasing a benchmark where prosthesis status is explicitly visible, future work will take people with limb differences as a target in pose estimation, rather than an afterthought, and will develop prosthesis-aware models on top of the signals that InclusiveVidPose provides.
> >
> > ---
> >
> > ### E1. Ethics and legal compliance
> >
> > > Concern about the legal and ethical implications of using YouTube and other web videos as sources for the dataset.
> > >
> >
> > **Response.** We follow fair use and fair dealing principles within the relevant jurisdiction(s) where the data are accessed and processed, and we rely on institutional copyright guidance. For completeness, we refer to both United States and Australian copyright law, which each provide limited exceptions for research and study. In the United States, Section 107 of the Copyright Act codifies the fair use doctrine and lists criticism, comment, news reporting, teaching, scholarship, and research as illustrative purposes that may qualify as fair use in appropriate circumstances. In Australia, the Copyright Act 1968 contains fair dealing provisions for research and study that permit the use of copyrighted material for genuine research when the dealing is fair and reasonable. In line with these frameworks, and as summarized in Section G of the supplementary material, we only access videos from lawful online platforms, we do not redistribute any video frames or re-encoded content, and we release only derived annotations and metadata. This design keeps the original media under the control and terms of the hosting platforms and rights holders while aligning our use of publicly available video with established research and study exceptions.

---

### Author Response · Authors · 2025-12-03
**Summary**

**Thank you for your work during this round.**

We sincerely thank you for stepping in during the unusually disrupted discussion phase and for acting as the final safeguard for this submission. We also appreciate the program chairs for their prompt and decisive handling of the situation and for trying to restore a fair process despite the broader instability in this ICLR cycle.

---

### Overall review picture

- All three reviewers provide positive and constructive assessments. Two reviewers give a rating of 8 and one gives 6, for an average rating of 7.33.
- All of them agree that **InclusiveVidPose** fills a clear gap by providing the first large-scale video based pose dataset that centers individuals with limb deficiencies and by introducing the LiCC metric for confidence consistency under missing or residual limbs.

**Commonly highlighted strengths**

- The problem is clearly motivated and addresses an underserved but clinically and socially important population with limb differences.
- The dataset is large, carefully curated, and video based, which allows annotators to disambiguate occlusion from true absence. Each frame includes keypoints, segmentation masks, boxes, tracking IDs, and per limb prosthesis status, making it broadly useful across tasks.
- The extended 25 keypoint schema with residual limb endpoints is technically sound and provides a concrete way to model altered anatomy rather than only intact skeletons.
- LiCC is viewed as a meaningful and practical metric that captures anatomical plausibility and confidence calibration, beyond standard AP and AR.
- The experimental section is comprehensive, covering many recent 2D pose estimators and reporting both COCO metrics and LiCC, which clearly show that current methods struggle on limb differences and that residual endpoints are harder to localize than standard joints.
- Reviewers consistently describe the paper as well written and easy to follow.

---

### Limited discussion opportunity

Because of this year’s disruption to the discussion phase, only one reviewer posts follow-up questions after the initial reviews. We provide detailed, point-by-point responses to all reviewers, but there is no chance to have further back-and-forth or to see updated comments or ratings from most of them.

---

### How we address questions and weaknesses in the revision

In the rebuttal and revision we respond point by point to the reviewers’ questions and weaknesses and adjust the paper accordingly:

- **Annotation quality and consistency.** We add quantitative analyses that separate the 8 residual endpoints from the 17 standard COCO joints, showing that standard joints are easier and benefit more from COCO while residual endpoints remain the main challenge.
- **Use of COCO pretraining and catastrophic forgetting.** We make explicit that all backbones are pretrained on COCO before training on InclusiveVidPose, and we highlight that the above analyses indicate no catastrophic forgetting on intact joints, while residual endpoints remain the main challenge.
- **Scope and evaluation protocol.** We sharpen the positioning as a dataset and benchmark plus metric contribution, explain the choice of single-frame evaluation, and include a video-oriented baseline as a reference.
- **Ethics and licensing.** We expand the ethics and licensing sections with explicit HREC approval, copyright guidance, and a Data Use Agreement that restricts use to non-commercial research and prohibits re-identification.
- **Minor issues.** We fix the missing “annotation team training” field and other small textual errors.

---

### Closing remark

In summary, the reviewers view this work as a solid dataset and benchmark contribution that addresses an important fairness gap in human pose estimation and that provides both a new resource and a new metric for the community. The revision directly answers the raised questions and strengthens the technical, empirical, and ethics sections.

We are grateful for your efforts in this challenging ICLR cycle and we hope this summary helps you in forming a final judgment about the paper.

---

### Meta-Review · Area_Chair_j2bk · 2026-01-06

**Summary:**

All reviewers gave positive ratings to this paper, appreciating the clear contributions of the proposed dataset and its advantages, which are supported by comprehensive experimental evaluations.

The major concerns raised by the reviewers are: (1) the lack of analysis of annotation quality and consistency; (2) the lack of utilization of temporal cues, despite the dataset being video-based; and (3) concerns regarding the dataset’s license.

In the authors’ rebuttal, all of the above concerns are sufficiently addressed.

**Reviewer Concerns:**

In the authors’ rebuttal, the reviewers’ concerns are well addressed, which are also clearly summarized in the authors’ final response. In particular, the authors provide quantitative analyses demonstrating annotation quality and consistency, explicitly resolve the licensing issue by obtaining the necessary approvals, and include additional experiments that apply multi-frame methods to their dataset.

**Reviewer Scores:**

Overall, the AC believes that the authors’ responses are convincing and clear. Given that all reviewers already gave positive ratings, the ratings are expected to remain unchanged.

---

### Decision · Program_Chairs · 2026-01-26

Accept (Poster)